# Searching Latent Program Spaces

**Matthew V. Macfarlane**[*][1]               **Clément Bonnet**[*]

[1]University of Amsterdam        [*]Equal contribution

## Abstract

General intelligence requires systems that acquire new skills efficiently and generalize beyond their training distributions. Although program synthesis approaches have strong generalization power, they face scaling issues due to the large combinatorial spaces that quickly render them impractical, requiring human-generated DSLs or pre-trained priors to narrow this search space. On the other hand, deep learning methods have had high successes, but they lack structured test-time adaptation and rely on heavy stochastic sampling or expensive gradient updates for fine-tuning. In this work, we propose the Latent Program Network (LPN), a novel architecture that builds in test-time search directly into neural models. LPN learns a latent space of implicit programs—neurally mapping inputs to outputs—through which it can search using gradients at test time. LPN combines the adaptability of symbolic approaches and the scalability of neural methods. It searches through a compact latent space at test time and bypasses the need for pre-defined domain-specific languages. On a range of programming-by-examples tasks, LPN either outperforms or matches performance compared to in-context learning and test-time training methods. Tested on the ARC-AGI benchmark, we demonstrate that LPN can both learn a compact program space and search through it at test time to adapt to novel tasks. LPN doubles its performance on out-of-distribution tasks when test-time search is switched on.

## 1 Introduction

The central goal of artificial intelligence has long been to create generally intelligent systems with human-like cognitive capabilities. While recent years have seen remarkable achievements in narrow AI domains, with systems achieving superhuman performance in games [Campbell et al., 2002, Silver et al., 2017, Vinyals et al., 2019] and specialized tasks [He et al., 2015, Jumper et al., 2021], we face a fundamental challenge: our systems struggle to generalize beyond their training distribution [Yu et al., 2024a] or to effectively adapt to novelty [Zhang et al., 2021]. This highlights a critical gap between optimizing for task-specific performance and achieving true general intelligence.

In *On the Measure of Intelligence*, Chollet [2019] argues that traditional benchmarks that measure skills alone are insufficient for developing generally intelligent systems. Such benchmarks can be "gamed" through either unlimited prior knowledge of the task encoded by developers or massive amounts of training data, masking a system's true ability to generalize and adapt. Instead, it is argued we need to measure skill-acquisition efficiency—how effectively a system can learn new tasks with minimal experience—and evaluate generalization capability in a way that controls for both prior knowledge and training data.

By controlling for prior knowledge and experience, the Abstraction and Reasoning Corpus (ARC-AGI) [Chollet, 2019] is a benchmark that measures skill acquisition efficiency rather than pure skills.

---

Correspondence to `<matthew.v.m@live.co.uk>`, `<clement.bonnet16@gmail.com>`

39th Conference on Neural Information Processing Systems (NeurIPS 2025).

Its few-shot learning setup requires systems to adapt to novel tasks with very limited data, highlighting learning efficiency. Importantly, ARC-AGI is designed to evaluate artificial intelligence systems while calibrated to human performance. Therefore, it serves as a useful compass when comparing general intelligence between humans and machines.

Current approaches to solving ARC-AGI and program synthesis benchmarks generally fall into two categories: inductive and transductive methods. (1) Inductive approaches [Parisotto et al., 2016, Devlin et al., 2017b, Butt et al., 2024] infer underlying programs from examples by generating explicit programs using a domain-specific language (DSL). However, handcrafting problem-specific DSLs is inherently unscalable for real-world applications and relies on the unrealistic assumption that a human-generated DSL is always provided. Progress on AI-generated DSLs[Ellis et al., 2021] has also been limited. In this work, we investigate methods that solve problems using only input-output example data. (2) In-context learning—also called transductive—methods [Devlin et al., 2017a, Kolev et al., 2020, Li et al., 2024a] train neural models to condition on a few examples and produce the desired output for a new input, implicitly inferring the program within the model's activations. These methods benefit from greater scalability by removing the need for a predefined DSL. However, in-context learning struggles with *consistency* [Devlin et al., 2017b], i.e., being able to map the inputs to the outputs of the specification itself. Performing fine-tuning at test time is a powerful way to resolve consistency, but happens to be extremely costly on large models and is also prone to overfitting given limited data [Li et al., 2021]. Additionally, in-context learning in transformers [Vaswani et al., 2017] suffers from the quadratic complexity of self-attention [Hübotter et al., 2024], which hinders the practicality of scaling specification size.

As an attempt to get the best of both inductive and transductive worlds, we introduce the Latent Program Network (LPN), which integrates the benefits of a scalable neural architecture with program induction. By representing implicit programs in a continuous latent space, LPN allows for efficient test-time program search. LPN is more efficient than test-time fine-tuning methods like in Devlin et al. [2017a] because it performs backpropagation only through a fraction of the total parameters. Also, by performing latent program aggregation and recombination in latent space, LPN removes the quadratic cost of attention when scaling specification size.

Our contributions are as follows. (1) We introduce a new architecture named the Latent Program Network (LPN) that builds in test-time adaptation by learning a latent space of programs and searching for the best latent representation given new data. (2) We show that gradient-based latent search during training optimizes the latent space for effective test-time adaptation, yielding significant performance improvements. (3) We demonstrate that LPN generalizes to specification sizes beyond those seen during training, even improving performance, unlike in-context learning, which fails to generalize without parameter fine-tuning at large specification sizes.

## 2 Related Work

The challenge of navigating vast program spaces has driven diverse approaches to program synthesis. Early researchers focused on deductive methods, using theorem-proving techniques to construct provably correct programs based on formal specifications [Manna and Waldinger, 1980]. However, the difficulty of obtaining formal specifications led to a shift toward inductive approaches [Solomonoff, 1964]. These methods instead infer programs from input-output examples [Shaw et al., 1975, Summers, 1977, Biermann, 1978], making program synthesis more practical for real-world applications.

However, for real-world problems, the exponential search space is a significant challenge for inductive program synthesis methods [Lee et al., 2018]. Three distinct paradigms have emerged to narrow down the search space, all leveraging learning. (1) Researchers have developed differentiable programming languages that integrate symbolic reasoning with continuous optimization [Feser et al., 2016, 2017, Gaunt et al., 2017]. (2) Neural networks have been trained as priors to constrain the search space and enhance the efficiency of program search techniques [Balog et al., 2016, Devlin et al., 2017b]. In this paradigm of leveraging neural priors for search, large language models (LLMs) have been demonstrated to be valuable for navigating complex search spaces, including discrete program synthesis [Wang et al., 2024, Li et al., 2024d, Barke et al., 2024]. CodeIt [Butt et al., 2024] leverages the pre-trained CodeT5 model [Wang et al., 2023] to guide discrete program search. (3) In-context/transductive approaches bypass intermediate program representations entirely

by (meta-) training neural networks to directly map input to output examples and a new input to its corresponding output [Devlin et al., 2017a,b, Kolev et al., 2020]. Neural networks offer a fully differentiable program representation, allowing program induction through gradient descent in parameter space, where network weights encode the program structure [Graves, 2014, Zaremba and Sutskever, 2014, Neelakantan et al., 2015, Kaiser and Sutskever, 2015, Kurach et al., 2015]. While early research focused on learning individual programs, more recent works have broadened the scope to learning multiple programs [Devlin et al., 2017a, Kolev et al., 2020, Li et al., 2024a]. Pre-trained LLMs can also be used to perform transductive reasoning, where they directly generate outputs from specifications included in the prompt—a process known as in-context learning [Gendron et al., 2023, Mitchell et al., 2023, Li et al., 2024a, Brown et al., 2020]. Recent works by Hendel et al. [2023], Yang et al. [2024] demonstrate the emergence of task vectors within LLM activations. LPN advances this approach by explicitly learning and disentangling the program space from other activation information. This separation yields two benefits: enhanced program representations and more efficient program space exploration. While such neural approaches eliminate the need for formal programming languages or DSLs, they sacrifice both interpretability and the ability to perform systematic search during inference. As Devlin et al. [2017b] demonstrated, neural approaches struggle to maintain *consistency* with given specifications and underperform compared to neural-symbolic methods [Parisotto et al., 2016].

Recent work has explored fine-tuning model parameters at test time—test-time training (TTT)—to resolve this inability to be consistent with the specification [Devlin et al., 2017a, Hottung et al., 2021b, Hübotter et al., 2024, Li et al., 2024a, Akyürek et al., 2024]. TTT treats the network parameters as a program, where test-time fine-tuning becomes a search through program space via parameter optimization. Since neural networks are universal function approximators [Hornik et al., 1989], the target program is likely to exist somewhere in this parameter space. Meta-learning approaches such as MAML [Finn et al., 2017], perform gradient updates on few-shot data in the full parameter space. LEO [Rusu et al., 2018] similar to our work explore a version of MAML, introducing a bottleneck, such that the inner loop optimisation is performed in latent space. Their decoder is then a hypernetwork [Ha et al., 2016] which conditions on the latent to generate the raw parameters for inference. LPN removes the need for such a hypernetwork leveraging the in-context learning abilities of transformers to directly perform generation conditioned on the latent variable.

Conditioning neural models on latent spaces has emerged as a powerful approach across diverse domains. Latent space optimization has been applied to molecule generation [Gómez-Bombarelli et al., 2018] and combinatorial optimization challenges, including the traveling salesman problem [Hottung et al., 2021a, Chalumeau et al., 2023]. Recent work has extended latent-based approaches to black-box optimization through energy-based models [Yu et al., 2024b], and symbolic mathematics, where latent optimization helps balance equation complexity with accuracy [Meidani et al., 2023]. In program synthesis, LEAPS [Trivedi et al., 2021] demonstrated how reinforcement learning could search latent program spaces for Karel programs [Pattis, 1994].

LPN can be interpreted through the lens of semi-amortised variational inference (SVI) [Kim et al., 2018, Marino et al., 2018]. SVI operates in two phases: first, it performs amortized variational inference using an encoder trained on the complete dataset. Then, for each data point, it performs additional updates to minimize the amortization gap [Gershman and Goodman, 2014]—the discrepancy between the log-likelihood and the evidence lower bound (ELBO) [Krishnan et al., 2018, Cremer et al., 2018]. The LPN equivalent of these updates for test examples is performing gradient descent in the latent space.

## 3 Background

**Program Synthesis** aims to generate deterministic programs in a target language, such that outputs generated from inputs are consistent with the given specification. Typically, the problem space $Y$ consists of programs formulated within a domain-specific language (DSL). Each task is defined by a specification set $X$, where each specification, $X_m \in X$, is described by a set of input/output (I/O) examples:

$$X_m = \{(x_1^m, y_1^m), \ldots, (x_n^m, y_n^m)\} \tag{1}$$

A program $f \in Y$ is considered to solve the task associated with $X_m$ if it satisfies: $\forall j \in [1, n], \quad f(x_j^m) = y_j^m$. This definition requires that the program replicates the output for each input in the specification. We denote $F_m$ to represent the true function that generates the I/O pairs.

**Program Synthesis Generalization.**   We consider the problem of applying a learned program to a new input rather than explaining the specification. Given a set of input-output examples generated by a program $F_m$ (not necessarily constrained to a DSL), along with an additional input $x_{n+1}^m$,

$$P_m = \{(x_1^m, y_1^m), \ldots, (x_n^m, y_n^m), x_{n+1}^m\}. \tag{2}$$

The objective is to generalize from the provided examples and predict the corresponding output for $x_{n+1}^m$. This can be done via induction or transduction [Li et al., 2024a]. If we do not limit the Kolmogorov complexity of programs [Kolmogorov, 1965, Solomonoff, 1964], we can find an explanation for any given specification, whether or not it corresponds to the true program that underlies the specification. Evaluating generalization performance not only tests the ability to explain a specification but also whether it can successfully infer the underlying program in its generality and apply it to a new input. This problem bears a resemblance to few-shot learning, with the difference of having only one task. The ARC-AGI benchmark [Chollet, 2019] falls under this program synthesis formulation. Each of its tasks is composed of input-output pairs represented as 2D grid of shape up to 30x30, whose cells can take any of 10 colors.

# 4   Latent Program Network (LPN)

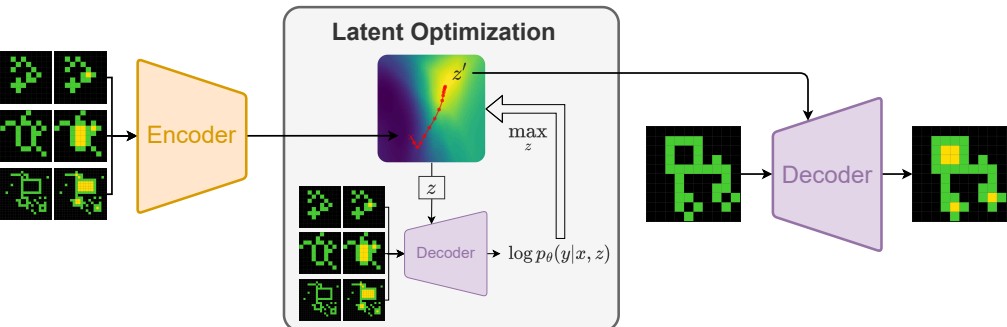

Figure 1: Inference of the Latent Program Network (LPN) model. (Left): the encoder maps I/O pairs to a latent space of encoded programs. (Middle): the latent program is refined during an optimization process to best explain the given I/O pairs (figure detailed in the appendix at Figure 20). (Right): the decoder executes the latent program to generate the desired output for a newly given input.

Prior work tackling programming by example with a neural approach has focused on directly training models to maximize the likelihood of decoding the correct output given a specification (in context) [Kolev et al., 2020]. However, we diverge from such transduction-based methods, as they cannot inherently adapt at test time or ensure specification consistency. Instead, we explicitly factorize inference into three core components visualized in Figure 1. First, we introduce a bottleneck that encourages the network to learn an explicit representation of programs via a compact latent space, an architecture also used in Neural Processes [Garnelo et al., 2018]. Secondly, we introduce a method for searching this latent space to explain the given data effectively. Lastly, conditioned on the latent program and a new input, we predict the output. Despite this added structure, the Latent Program Network (LPN) remains fully differentiable end-to-end. This structure provides several key benefits. Firstly, explicitly learning program representations acts as a good neural network prior for generalization. Secondly, we can use this program encoding to verify that our initial guess for a latent program explains the given data. If not, the program latent can be refined at test-time to best explain the given data. Lastly, LPN removes issues faced by transductive approaches that overfit to specification sizes (number of input-output pairs) seen during training.

## 4.1   Latent Program Inference

LPN is composed of three core architectural components to perform inference. A neural encoder, a neural decoder, and a latent space optimization. In this section, we discuss each component and then outline how to train the full system in later sections.

**Encoder.** The probabilistic encoder is trained to approximate the Bayesian posterior over program latents. Specifically, it maps an input-output pair $(x, y)$ to a distribution in the latent space $q_\phi(z|x, y)$, representing possible programs that could explain the given input-output mapping. Using a variational approach is important because, for any given input-output pair, there exists a broad range of possible programs that map the input to the output, even when restricting to, e.g., programs of low Kolmogorov complexity [Solomonoff, 1964, Kolmogorov, 1965]. We discuss LPN as semi-amortised variational inference in Section D. Intuitively, the encoder is trained to learn an abstract representation of programs in a continuous latent space by implicitly encoding input-output pair examples. In practice, we use a multivariate normal distribution whose mean $\mu$ and diagonal covariance $\Sigma$ parameters are inferred by the encoder. To take advantage of hardware parallelization, the encoder can process all the I/O pairs in a given specification in parallel. By encoding each pair independently and aggregating using the mean LPN is permutation invariant to the specification order, as opposed to a naive sequence model that would concatenate I/O pairs.

**Decoder.** The probabilistic decoder is responsible for mapping a latent program and an input to its expected corresponding output, directly predicting the output pixel-by-pixel instead of via a DSL. It models the distribution of possible outputs $y$ given an input $x$ and a latent $z$. Note that even if the underlying I/O mappings are deterministic, we still use a probabilistic decoding framework $p_\theta(y|x, z)$ to be compatible with maximum likelihood learning. Figure 20 shows the decoder generating different outputs by keeping the input fixed but varying the latent program, which in this figure represents a specific grid pattern to reproduce. In a real task, the aim of this encoder-decoder system is to learn a compressed representation of the space of possible programs we care about (e.g., in the case of ARC-AGI, this would correspond to programs that use the Core Knowledge priors [Chollet, 2019]).

**Latent Optimization.** The encoder is trained to approximate the posterior over programs and may not encode the right abstraction given an I/O pair. Especially if the task is very novel, the encoder may fail at producing the right latent program, which, fed to the decoder, would generate the wrong output. Therefore, we include a middle stage of latent optimization where, starting from the encoder's prediction $z$, we search for a better latent program $z'$, one that would better explain the observed data according to the decoder $p_\theta$. The search process is generally denoted $z' = f(p_\theta, z, x, y)$ and can be implemented in several ways (c.f. section 4.2). Analogous to system 1 / system 2 thinking [Kahneman, 2011], we can think of the encoder generating an intuitive first guess as to what the observed program may be (system 1), and the latent optimization process executing a search for hypotheses that would better explain the observations (system 2). See Section C for test-time inference pseudo-code.

$$
\begin{array}{ccc}
\text{Encoder} & \text{Latent Optimization} & \text{Decoder} \\
z \sim q_\phi(z|x, y) & z' = f(p_\theta, z, x, y) & \hat{y} \sim p_\theta(y|x, z')
\end{array} \tag{3}
$$

## 4.2 Search Methods for Latent Optimization

Given $n$ input-output pairs $\{(x_i, y_i)\}_{i=1...n}$, the search process $z' = f(p_\theta, z, x, y)$ attempts to find a $z'$ that satisfies:

$$
z' \in \arg\max_z \sum_{i=1}^{n} \log p_\theta(y_i|x_i, z) \tag{4}
$$

This means we search for the latent that would most likely make the decoder generate the right outputs given the corresponding inputs. By finding a latent that can explain all the input-output pairs, the latent solution to the optimization problem is more likely to generalize to a new input-output pair. We describe here two instantiations of the search process, namely a *sampling* and a *gradient ascent* algorithm, both acting in the latent space of programs. We leave for future work the exploration of other search methods like evolutionary strategies [Hansen and Ostermeier, 2001, Chalumeau et al., 2023] that could better trade-off exploration and exploitation of the latent space.

**Sampling.** A naive version of the latent search process is to sample a batch of latents from either the prior distribution $p(z)$ or around the approximate Bayesian posterior $q_\phi(z|x_i, y_i)$ and select the latent that gives the highest log likelihood of decoding the given input-output pairs. Specifically, for all $k \in [1, K]$, $z_k \sim p(z)$, and we select $z' \in \arg\max_{z_k} \sum_{i=1}^{n} \log p_\theta(y_i|x_i, z_k)$. *Sampling* asymptotically converges to the true maximum-likelihood latent (equation 4) and can prove useful

when the function to optimize (here, the decoder log-likelihood) is not differentiable. However, the efficiency of sampling-based search decreases exponentially with the dimension of the latent space, which makes it impractical for most applications.

**Gradient Ascent.** Since the decoder is a differentiable neural network, its log-likelihood $\log p_\theta(y|x, z)$ is also differentiable with respect to $z$ and one can use first-order methods like gradient-ascent to efficiently search through the latent space for a solution to the latent optimization problem (equation 4, see Figure 20 for a visualization of a 2D latent space trained on grid patterns). This visualization highlights that only a small portion of the latent space can explain all the input-output pairs, corresponding to high decoding likelihood. Notably, poor initialization can lead the search to converge to different local minima, highlighting the importance of amortized inference from the encoder. We initialize $z'_0$ as the mean of all the pair latents sampled from the encoder, and refine it iteratively as follows:

$$\forall k \in [1, K], \quad z'_k = z'_{k-1} + \alpha \cdot \nabla_z \sum_{i=1}^{n} \log p_\theta(y_i \mid x_i, z)\big|_{z=z'_{k-1}} \tag{5}$$

The series $(z'_k)_{k \in [1,K]}$ should exhibit increasing decoding likelihood if the step size $\alpha$ is small enough. In practice, we generate the output with the best latent found during the gradient ascent algorithm, which may not always be the one that is obtained after taking the last gradient step ($z'_K$).

## 4.3 Training

To train the LPN system end-to-end, we assume we have a dataset of tasks, where a task is defined as $n$ input-output pairs $(x_i, y_i)$ generated by the same program. To simulate the test conditions of predicting a new input from a given specification, we design the training procedure to reconstruct each of the outputs $y_i$ from their inputs $x_i$ and all the $n-1$ other pairs $(x_j, y_j)_{j \neq i}$. We emphasize that we do not use the specific pair $(x_i, y_i)$ to reconstruct $y_i$, which would lead to the encoder directly compressing the output $y_i$ as a shortcut without learning program-related abstractions.

When reconstructing output $y_i$, we first sample latents $z_j$ from the encoder $q_\phi(z|x_j, y_j)$ for all $j \neq i$. We then aggregate them by computing their mean $\frac{1}{n-1} \sum_{j \neq i} z_j$, then we perform latent optimization using e.g., gradient ascent to obtain $z'_i$. Finally, we compute the negative log-likelihood of the right output $y_i$ using its corresponding input $x_i$ and the refined latent $z'_i$. In practice, we compute the cross-entropy loss of the decoder logits $p_\theta(\hat{y}_i|x_i, z'_i)$ and the labels $y_i$, which is derived from maximizing the likelihood of a categorical distribution. The full training pipeline is detailed in Section C. Specifically, we compute the reconstruction loss $\mathcal{L}_{\text{rec}}$ and the KL loss $\mathcal{L}_{\text{KL}}$ between the approximate posterior and the prior:

$$\mathcal{L}_{\text{rec}}(\phi, \theta) = \sum_{i=1}^{n} -\log p_\theta(y_i|x_i, z'_i) \qquad \mathcal{L}_{\text{KL}}(\phi) = \sum_{i=1}^{n} D_{\text{KL}}\left(q_\phi(z|x_i, y_i) \parallel \mathcal{N}(0, I)\right) \tag{6}$$

The dependence of the reconstruction loss $\mathcal{L}_{\text{rec}}(\phi, \theta)$ in $\phi$ arises from using the reparameterization trick [Kingma, 2013] when sampling each latent $z_j$. Indeed, we first sample a normal random vector $\epsilon_j \sim \mathcal{N}(0, I)$, then we infer the mean $\mu_j$ and diagonal covariance $\Sigma_j$ using the encoder and recompose the latent $z_j = \mu_j + \epsilon_j \cdot \Sigma_j$. Then, $z'_i$ is used by the decoder to reconstruct the output. Note that we can decide whether to let the decoder gradient flow through the latent update. Indeed, it is more computationally efficient to stop the gradient through the update, by changing line 9 of algorithm 2 with $z'_i = z'_i + \alpha \cdot \overline{g'_i}$, where $g'_i = \nabla_z \sum_{j \neq i} \log p_\theta(y_j|x_j, z)|_{z=z'_i}$, with $\overline{g'_i}$ noting a stop-gradient on $g'_i$.

We denote $\beta$ as the weighting factor that balances the reconstruction and KL terms [Burgess et al., 2018], which gives the combined training objective: $\mathcal{L}_{\text{total}}(\phi, \theta) = \mathcal{L}_{\text{rec}}(\phi, \theta) + \beta \mathcal{L}_{\text{KL}}(\phi)$. This training procedure offers some freedom in the latent optimization, i.e., how to compute $z'_i$ from $z_i$. Training with gradient ascent latent optimization (as detailed in Algorithm 2) incurs a compute overhead due to the cost of the latent gradient computation through the decoder. Although we may use a high compute budget at test-time, we propose to use a small number of gradient ascent steps during training, ranging from 0 to 5 steps.

# 5 Experiments

| Training | Inference | | | | | |
|----------|-----------|---|---|---|---|---|
| | Grad 0 | Grad 1 | Grad 5 | Grad 20 | Grad 100 | Sample 250 |
| Grad 0 | 3.2 (2.7) | 3.6 (3.0) | 18.8 (14.4) | 52.5 (25.0) | 67.5 (20.0) | 3.2 (2.7) |
| Grad 1 | **8.6 (4.4)** | **44.6** (10.9) | **85.4** (7.6) | **98.4** (1.4) | **99.5** (0.5) | **10.2 (5.3)** |
| Grad 1 ** | 0.6 (0.1) | 13.7 (3.0) | 60.2 (7.5) | 88.9 (6.0) | 94.1 (3.8) | 0.7 (0.2) |
| Grad 5 | 0.0 (0.0) | 0.4 (0.3) | 31.9 (11.2) | 88.5 (11.9) | 98.1 (2.1) | 0.5 (0.4) |
| Sample 5 | 6.1 (4.4) | 8.2 (6.5) | 27.7 (21.6) | 56.3 (27.5) | 72.2 (21.2) | 6.1 (4.4) |

Table 1: Ablation of LPN training and inference methods on the *Pattern* task, reporting exact match accuracy (%). Rows/columns represent different training/inference methods, differing only in the latent optimization. *Grad [N]* stands for $N$ gradient ascent steps, *Sample [N]* for $N$ samples from the encoder distribution without leveraging any gradients, and *Grad 1* ** means that the decoder parameter gradient flows through the latent optimization update. Training was performed for 20k steps with 3 seeds, aggregating performance as mean (and standard deviation in brackets) over the 3 runs. Bold values indicate the best training method for each inference regime. See expanded table in Section B.1.

## 5.1 Setup

We consider the ARC-AGI 2024 challenge [Chollet et al., 2024] as the testing domain for our method. This program synthesis benchmark encompasses diverse tasks designed to test adaptation and out-of-distribution generalization, rather than memorization. We implement both the LPN encoder and decoder as small transformers [Vaswani et al., 2017], see Section G for full architecture details. We introduce the simpler *Pattern* task to investigate LPN's dynamics before large-scale ARC-AGI training, see Section A. It generates 10x10 black input grids with a blue pixel indicating where a 4x4 program-specific pattern should be pasted. This task is sufficient to demonstrate weaknesses in deep learning models that do not leverage test-time computation (see Section 5.5).

**Baselines.** We compare LPN to an in-context learning method Kolev et al. [2020], Li et al. [2024a], which encodes each of the input-output pairs and then concatenates these embeddings to condition output prediction, notably never producing any intermediate program embeddings. We discuss the motivation for this design in Section H. Then, we also compare to test-time fine-tuning [Devlin et al., 2017a, Akyürek et al., 2024], which, given a test-time specification, performs parameter-based gradient updates on the in-context model. See Section H for details on baseline implementations and Section E for hyperparameters.

## 5.2 Pattern Task

We compare a variety of LPN training and inference methods in Table 1, to better understand the dynamics of LPN. We aim to first answer whether LPN can self-improve at test-time by searching its latent space, and then how the training inference strategy affects performance. For each training method, we train a small 1M-parameter model for 20k steps with a batch size of 128 and evaluate it with different inference modes. We find that inference using no test-time adaptation performs poorly across all instances of LPN (and in-context baselines, see Section B.2). This shows that amortizing inference is difficult for models of this size and training time.

All variations of LPN training show strong scaling in performance as the number of gradient steps at test time is scaled. This demonstrates that LPN is capable of test-time adaptation to improve pre-trained amortization performance. We visualise the decoded outputs at many points in the latent space of LPN on a simple pattern task in Section B.4 showing structure in the latent program representations, and Section B.4 shows the likelihood of the specification is smooth across the space, enabling gradient based test time search.

When comparing different LPN training inference strategies, 1 gradient ascent step of latent optimization shows higher returns than training with mean pooling, with the difference particularly pronounced when scaling the inference budget. With 100 steps of gradient ascent at test time,

LPN training with *Grad 0* gets an accuracy of 67.5%, whereas training with one gradient step (*Grad 1*) reaches 99.5%. This demonstrates the benefits of training the latent space with the awareness that gradient ascent will be performed at test-time, an important inductive bias for the LPN architecture. We also observe that gradient ascent vastly outperforms sampling-based search, validating that a search without any gradient signal is highly inefficient. Lastly, we perform training with and without the stop gradient on the gradient update itself. We found that using the stop gradient actually leads to higher test time performance while also being more computationally efficient.

## 5.3 String Manipulation Task

To investigate the generalizability of our results beyond environments with significant spatial structure, we perform an ablation on a synthetic sequence task and replicate the analysis previously conducted on the pattern dataset.

This synthetic dataset features a vast program space, with over 100 million unique programs, each defined by composing 3 to 5 parameterized rules that transform sequences of numbers (ranging from 0 to 4), see Section B.7 for further details. Our experiments demonstrate that Test-Time Training (*TTT*) exhibits overfitting, as evidenced by its performance degradation when gradient fine-tuning is performed. In contrast, LPN maintains robust performance without overfitting. Furthermore, incorporating a single gradient step during training enhances LPN performance marginally at higher inference gradients.

|  | | Inference | |
|---|---|---|---|
| Training | Grad 0 | Grad 10 | Grad 100 |
| In-Context | 77.7 (1.2) | - | - |
| TTT | **78.1** (1.1) | 72.8 (0.8) | 11.8 (1.9) |
| LPN Grad 0 | 75.6 (1.7) | 83.3 (0.8) | 81.3 (0.5) |
| LPN Grad 1 | 71.3 (0.4) | **86.1** (1.5) | **81.8** (1.8) |
| LPN Grad 1 ** | 67.9 (1.3) | 85.0 (0.4) | 81.0 (1.3) |

Table 2: Exact match accuracy (%) on the test set for the string manipulation task. The comparison includes an *In-Context* baseline, Test-Time Training (*TTT*) ($lr = 1e-4$), and three *LPN* variants (with $lr = 1e-1$): *Grad 0*, *Grad 1*, and *Grad 1* ** (decoder gradient flows through latent optimization). Performance is averaged over 3 runs, with standard deviations in parentheses.

## 5.4 Starting test-time search from the encoder

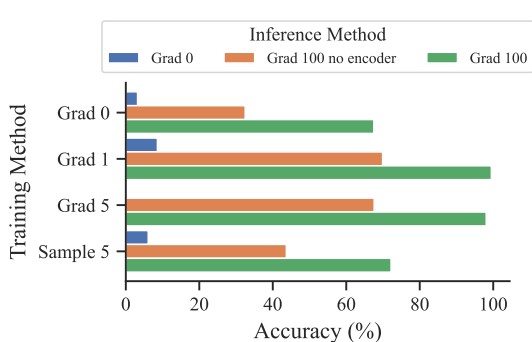

Figure 2: Ablation on the role of the encoder and latent optimization. Latents are initialized from the mean of the encoder latents (except for orange). *Grad N* stands for doing latent optimization with $N$ gradient steps. Both the encoder initialization and the latent optimization matter for LPN.

We experiment on the *Pattern* task again, this time to assess the benefits of amortizing inference. We ablate the impact of initializing latent optimization with the encoder versus using the prior.

We specifically compare no latent search (blue) to doing latent optimization in two ways: sampling from the prior $z \sim p(z)$ (orange) and sampling from the encoder $z \sim q_\phi(z|x, y)$ (green).

Initializing latent search with the encoder is critical for performance across all training methods. This validates the intuition that LPN can perform fast system 1-like reasoning using the encoder and then narrow down the search space during latent optimization, simulating system 2 reasoning.

## 5.5 Adapting Out-Of-Distribution

A significant challenge to deep learning methods is generalizing to out-of-distribution (OOD) tasks. We study the OOD behavior of LPN, in-context learning, and parameter fine-tuning on the *Pattern* task in Table 3. In-context learning has no test time adaptation performance as it has no inherent mechanism to scale compute beyond a single forward pass. We include full experiments on a range of varying OOD settings in Section B.2.

When trained with 1 step of gradient ascent, LPN recovers high accuracy (88%) using 100 steps of gradient ascent at inference time. TTT and in-context learning are incapable of recovering the right OOD pattern at inference time. We perform a similar experiment in Section B.7 on a synthetic integer sequence task with less spatial structure, with the same conclusions.

|  | Inference | | |
|---|---|---|---|
| Training | Grad 0 | Grad 10 | Grad 100 |
| In-Context | 0.0 (0.0) | - | - |
| TTT | 0.0 (0.0) | 1.8 (0.7) | 0.3 (0.1) |
| LPN Grad 0 | **0.3** (0.5) | 18.8 (14.5) | 41.1 (29.6) |
| LPN Grad 1 | 0.0 (0.0) | **59.9** (11.6) | **88.0** (5.3) |
| LPN Grad 2 | 0.0 (0.0) | 38.5 (13.0) | 81.8 (10.9) |

Table 3: Out-of-distribution (OOD) performance on the *Pattern* task, measured as exact match accuracy (%). Performance is averaged over 3 training runs with different seeds, with standard deviation in parentheses.

## 5.6 Scaling Specification

For any learning method that can handle varying amounts of data, an important question is how the method scales to handling more data than it was trained on. In Figure 3 we evaluate in-context learning, TTT, and LPN with and without gradient search in the OOD setting of the *pattern* task, with varying specification sizes. LPN smoothly generalizes to different specification sizes, with only a small drop in performance with less data, and keeps improving with more data. In-context learning overfits to the specification size it is trained on, and TTT only performs well once given a sufficiently large specification and high inference compute. We ablate this in Section B.5, repeating the experiment for varying training specification sizes.

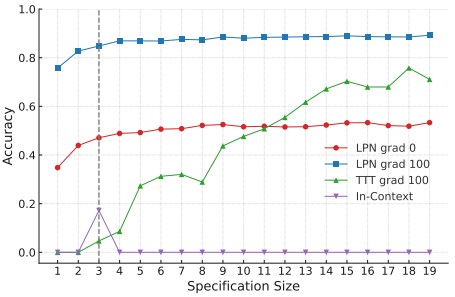

Figure 3: Exact match accuracy (%) on the out-of-distribution (OOD) *pattern* task as the specification size is scaled from 1 input output pair to 19 pairs, for different inference methods.

## 5.7 ARC-AGI 2024

**Setup**   We train on the `re-arc` dataset [Hodel, 2024], designed to be in distribution relative to the ARC training set (which we don't train on) Li et al. [2024b]. The evaluation set is significantly OOD, representing a challenging generalization experiment. We train a 178M-parameter LPN with a 256-dim latent space for 100k steps for 2 days on a TPU v4-32, see Section E. Leveraging insights from pattern task experiments, we use gradient ascent in LPN during training. To remain efficient, we train LPN in Grad 0 mode for 95k steps and then fine-tune in Grad 1 for a further 5k steps, which we find gives a marginal boost in performance at lower levels of test time compute Section B.8.

**Results**   Table 4 shows top-2 accuracy on ARC-AGI (standard to ensure resolvability of all problems). Even after significant amortization of inference, we observe large performance gains from scaling gradient search at test time. LPN significantly outperforms TTT in-distribution. We also see strong evidence that the LPN has learned a structured latent space for representing programs, demonstrated by a t-SNE plot of the embeddings of different programs in the training set, see Section F. Out of distribution, LPN doubles its performance by using latent space search at test-time (scaling FLOPs from 2e11 to 2e15). Using from 2E+11 to 2E+13 FLOPs, LPN outperforms TTT, until we scale test-time compute to much higher levels, where TTT achieves higher performance. This finding is likely due to the lack of diversity in programs seen by LPN during training, relative to experiments

| FLOPs | In-Distribution | | | | Out-of-Distribution | | | |
|---|---|---|---|---|---|---|---|---|
| | LPN | TTT | CodeIt | Mirchandani | LPN | TTT | CodeIt | Mirchandani |
| 2E+11 | 68.75 | 45.75 | - | - | 7.75 | 5.85 | - | - |
| 2E+12 | 75.95 | 51.75 | - | - | 10.25 | 7.35 | - | - |
| 2E+13 | 80.00 | 58.50 | - | - | 15.25 | 13.50 | - | - |
| 2E+14 | 76.25 | 58.75 | - | - | 15.50 | 16.00 | - | - |
| 2E+15 | 78.50 | 57.00 | - | - | 15.50 | 15.25 | - | - |
| » 2E+15 | - | - | - | 14.00 | - | - | 14.75 | 6.75 |

Table 4: Results on ARC-AGI, measured by exact match accuracy (%), for varying values of FLOPs. In-Distribution refers to puzzles in the ARC train set , not included in the training set, but are in-distribution relative to re-ARC. Out-of-distribution refers to puzzles from the eval set. LPN and TTT stand for Latent Program Network **(ours)**, and Test-Time Training. CodeIt and Mirchandani are baselines from prior work.

on other datasets, limiting the ability to have a smooth expressive latent space. See Section B.11 for an ablation on the performance contribution to LPN from encoder initialization and gradient search.

Relative to the pattern task, TTT is less susceptible to overfitting on ARC-AGI, possibly due to the use of larger model sizes. Both methods outperform two ARC-specific LLM approaches that leverage far larger pre-trained models, including CodeT5 [Butt et al., 2024] (220M parameters) and text-davinci [Mirchandani et al., 2023] (175B parameters). This work represents an improvement over such methods due to significantly lower inference and training cost, and greater generalization from a limited training distribution. At the time of writing, the highest performing model on ARC-AGI is o3-preview(low) (LLM) which achieves 75.7% at a cost of $200 per task. Both the training and inference costs are far beyond this work, there is an opportunity for future work to investigate scaling up this work to such a scale. To further contextualise the results compared to other methods applied to ARC-AGI we outline in Section I a complete table of leading and diverse approaches to the benchmark highlighting the training assumptions and compute for all methods where the information is public.

In section Section B.9 we investigate whether the problems solved, on the ARC-AGI evaluation test set, by TTT and LPN are different. We find that indeed different subsets of the test set are solved by each method and we visualize 3 examples of solutions only solved by LPN and TTT respectively.

Lastly, in Section B.10, we explore LPN's capacity for compositional generalization at test time. We find that the latent search process can compose programs seen during training into novel combinations, which avoids the overfitting seen in models without search, that tend to execute only a single program. However, this capability is not robust; our analysis identifies both successful examples where the search enables composition and clear failure cases where it does not.

# 6   Conclusion

We introduced Latent Program Network (LPN), a novel approach to inductive program synthesis leveraging continuous latent spaces for efficient test-time adaptation. LPN integrates adaptation directly into its architecture, refining latent representations through gradient-based search, which we identify as an efficient adaptation method. We show that LPN generalizes beyond the training distribution, relying on test-time adaptation rather than expanded synthetic datasets.

**Limitations and Future Work.**   A limitation of this work is the limited diversity of programs on which LPN is trained. While augmentations are used during training, the distribution of programs remains narrow and restricts the potential for learning an expressive complex latent space. We limit gradient-based search to standard optimizers. Future work could explore alternative optimization methods, such as evolution strategies, and explore discrete program representations to enhance compositional generalization.

## Acknowledgments and Disclosure of Funding

We thank Google's TPU Research Cloud (TRC) for supporting this research. We are also grateful to Nathan Grinsztajn, Natasha Butt, Levi Lelis, and Jessica Hu for their feedback on the early versions of the paper. Matthew Macfarlane is supported by the LIFT-project 019.011, which is partly financed by the Dutch Research Council (NWO). We also thank reviewers for their detailed feedback, which we have integrated into the final version of the paper.

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

# A  Datasets

## A.1  Pattern Task

The ARC-AGI challenge contains many diverse programs leveraging different knowledge priors. Injecting these priors into LPN by training the model to master ARC-like tasks requires significant compute resources when training from scratch, without an LLM-based initialization. Therefore, to investigate the training dynamics and properties of LPN before such a large-scale training, we develop a simpler task called *Pattern* task (see figure  4) within the same domain, but using a narrow distribution of pattern-like programs. This specific task always generates fully-black 10x10 inputs with a single blue pixel at a random location that defines where the output pastes a 4x4 pattern sampled from a uniform distribution. The pattern is program-specific, meaning it is the same across different pairs but varies from specification to specification. This task demonstrates how deep learning methods without test-time computation may still make errors on such tasks. We then extend this task to study an out-of-distribution setting in section 5.5.

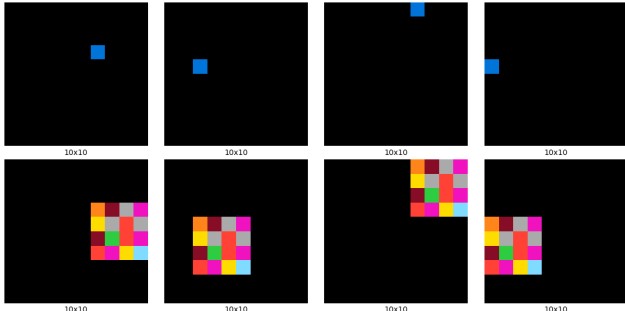

Figure 4: Example of input (top row) and output (bottom row) pairs of a specification sampled from the *Pattern* task. Each sample is a batch of 4 pairs that share the same pattern.

## A.2  ARC-AGI

We evaluate LPN on the ARC-AGI benchmark by training it on the `re-arc` dataset [Hodel, 2024],a dataset designed to be in distribution relative to the ARC training set, previously used for training large language models and ARC specific transformers to solve ARC-AGI Li et al. [2024b], Akyürek et al. [2024]. Most previous works however expand beyond this dataset, usually synthetically adding additional programs [Butt et al., 2024], or crafted datasets. We restrict training to tasks from only this dataset with the goal of assessing generalization instead of closing a performance gap with more data. This also ensures LPN solely learns from core knowledge priors without data leakage from the evaluation set. The evaluation set is known to be significantly out of distribution and therefore represents a challenging generalization experiment. We train a 178M-parameter LPN with a 256-dim latent space for 100k steps (batch size 256) for 2 days on a TPU v4-32, see Section E for further details, in terms of data, this amounts to 51M I/O pairs. Leveraging insights from smaller scale experiments we make use of gradient in LPN during training. We train LPN in Grad 0 mode for 95k steps and then fine tune in Grad 1 for a further 5k steps. We outline our hyperparameter procedure for both LPN and TTT for ARC-AGI in Section E.

# B Expanded Experiments

## B.1 LPN Training Extended

We provide an expanded version of Table 1 with additional ablations for both the training and inference axes.

| Training | Inference | | | | | |
|---|---|---|---|---|---|---|
| | Grad 0 | Grad 1 | Grad 5 | Grad 20 | Grad 100 | Sample 250 |
| Grad 0 | 3.2 (2.7) | 3.6 (3.0) | 18.8 (14.4) | 52.5 (25.0) | 67.5 (20.0) | 3.2 (2.7) |
| Grad 1 | 8.6 (4.4) | **44.6** (10.9) | **85.4** (7.6) | **98.4** (1.4) | **99.5** (0.5) | 10.2 (5.3) |
| Grad 1 ** | 0.6 (0.1) | 13.7 (3.0) | 60.2 (7.5) | 88.9 (6.0) | 94.1 (3.8) | 0.7 (0.2) |
| Grad 5 | 0.0 (0.0) | 0.4 (0.3) | 31.9 (11.2) | 88.5 (11.9) | 98.1 (2.1) | 0.5 (0.4) |
| Grad 5 ** | 0.0 (0.0) | 0.0 (0.0) | 9.9 (2.9) | 87.1 (6.0) | 95.1 (3.3) | 0.0 (0.0) |
| Sample 5 | 6.1 (4.4) | 8.2 (6.5) | 27.7 (21.6) | 56.3 (27.5) | 72.2 (21.2) | 6.1 (4.4) |
| Sample 25 | **10.8** (8.0) | 13.3 (10.1) | 39.9 (21.4) | 72.3 (18.5) | 87.9 (9.2) | **10.8** (8.0) |

Table 5: Ablation study of LPN training and inference methods on the *Pattern* task. Rows represent training methods: LPN *Grad [N]* for N gradient steps, LPN *Grad [N]* ** with gradient flow through latent optimization (analogous to meta-learning), and *Sample [N]* for N random samples. Columns represent inference methods: *Grad [N]* for N gradient ascent steps and *Sample [N]* for random search with N samples. Training was performed for 20k steps with 3 seeds, with performance reported as mean (standard deviation) over 3 runs. Bold values indicate the best performance per inference method.

## B.2 Adapting Out-Of-Distribution

We provide expanded results for the out-of-distribution (OOD) experiments outlined in Section 5.5. We include results for varying levels of OOD starting with in-distribution, weak OOD, and strong OOD.

| Training | Inference | | |
|---|---|---|---|
| | Grad 0 | Grad 10 | Grad 100 |
| In-Context | 15.3 (0.6) | - | - |
| Test Time Training | 15.3 (0.6) | 17.0 (1.7) | 0.78 (0.69) |
| LPN Grad 0 | **30.2** (15.7) | 72.8 (29.5) | 82.2 (21.3) |
| LPN Grad 1 | 26.6 (7.4) | **98.0** (0.6) | **99.2** (0.6) |
| LPN Grad 2 | 14.4 (3.5) | 97.2 (1.6) | 98.9 (1.2) |
| LPN Grad 3 | 1.0 (0.7) | 85.7 (6.3) | 98.3 (1.9) |
| LPN Grad 1 ** | 10.6 (6.4) | 93.8 (5.1) | 97.4 (1.9) |

In-distribution

| Training | Inference | | |
|---|---|---|---|
| | Grad 0 | Grad 10 | Grad 100 |
| In-Context | 2.1 (0.9) | - | - |
| Test Time Training | 2.1 (0.9) | 2.1 (1.1) | 0.00 (0.00) |
| LPN Grad 0 | **7.6** (5.6) | 51.3 (35.7) | 62.8 (38.3) |
| LPN Grad 1 | 7.4 (4.4) | **93.1** (4.3) | **97.7** (2.2) |
| LPN Grad 2 | 3.0 (2.6) | 86.1 (6.0) | 95.9 (2.1) |
| LPN Grad 3 | 0.0 (0.0) | 55.0 (7.8) | 93.9 (3.8) |
| LPN Grad 1 ** | 1.3 (1.0) | 81.7 (13.3) | 91.5 (5.5) |

Weakly out-of-distribution

| Training | Inference | | |
|---|---|---|---|
| | Grad 0 | Grad 10 | Grad 100 |
| In-Context | 0.0 (0.0) | - | - |
| Test Time Training | 0.0 (0.0) | 1.8 (0.7) | 0.3 (0.1) |
| LPN Grad 0 | **0.3** (0.5) | 18.8 (14.5) | 41.1 (29.6) |
| LPN Grad 1 | 0.0 (0.0) | **59.9** (11.6) | **88.0** (5.3) |
| LPN Grad 2 | 0.0 (0.0) | 38.5 (13.0) | 81.8 (10.9) |
| LPN Grad 3 | 0.0 (0.0) | 11.3 (9.3) | 72.0 (14.0) |
| LPN Grad 1 ** | 0.0 (0.0) | 40.9 (19.8) | 71.1 (14.3) |

Strongly out-of-distribution

Table 6: Study of the out-of-distribution (OOD) performance on the *Pattern* task. Models are trained on patterns that have a density of 50% (half black, half colored), then evaluated on the same distribution, on a density of 75% (weakly OOD) and 100% (strongly OOD). Performance is averaged over 3 training runs with different seeds, with standard deviation in parentheses.

## B.3 Validating the Decoder

Training deep networks from scratch to solve ARC-like tasks has been challenging [Li et al., 2024c]. If it is the case that neural networks struggle even to learn to execute single programs this represents a significant bottleneck to training models from scratch on a broad distribution of programs. Therefore, before training LPN end-to-end, we conclusively show that our decoder architecture does not suffer from such a bottleneck, and can learn individual programs.

We show 5 of the 400 total tasks from the ARC-AGI training set, and for each of these tasks, we train a small LPN architecture of 800k parameters (except for the last task which required a bigger model with 8.7M parameters) on the corresponding task generator from `re-arc` [Hodel, 2024]. Specifically, we select the first five tasks from the `arc-agi_training_challenges` json file (`007bbfb7`, `00d62c1b`, `017c7c7b`, `025d127b`, `045e512c`) shown in figure Figure 5.

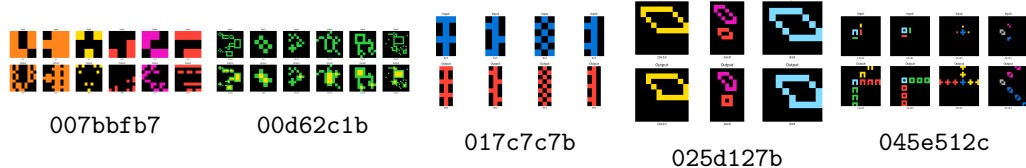

Figure 5: Overfit training on the first 5 ARC-AGI training tasks. The captions correspond to task IDs. For each task, the top row contains the input grids, and the bottom row the output grids. Each task consists of observing all pairs but the first and inferring the output of the leftmost pair given its input. Each curve corresponds to a separate training run.

We evaluate both the distribution of `re-arc` generated tasks on which it is trained and on the true task from the ARC-AGI training set in figure Figure 6. We show that for each task, the small LPN decoder-only model successfully learns individual programs and manages to solve the corresponding ARC-AGI task. Therefore, our model outperforms previously reported results in Li et al. [2024c], and concludes that our architecture does not suffer from a decoder bottleneck. Note that the encoder is not helpful in this experiment since the task is always the same. Our later results on ARC-AGI Section 5.7 take this a step further and show that we can learn a single transformer architecture capable of executing over 270 programs in the ARC training dataset.

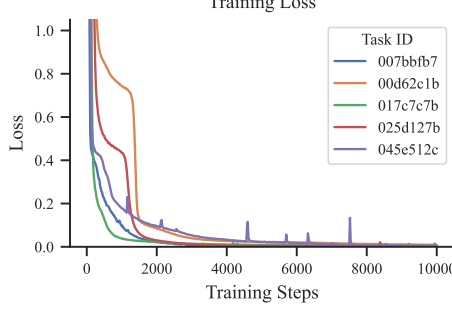

|  | Pixel Accuracy | |
| Task | `re-arc` | ARC-AGI |
| --- | --- | --- |
| `007bbfb7` | 100 | 100 |
| `00d62c1b` | 96.5 | 100 |
| `017c7c7b` | 99.7 | 100 |
| `025d127b` | 100 | 100 |
| `045e512c` | 98.3 | 100 |

(a) Training loss

(b) Performance after 10k steps of training

Figure 6: Training loss and performance of LPN training on 5 of the `re-arc` distributions. For each task, only samples from the `re-arc` generators are used for training. The corresponding ARC-AGI tasks are never seen during training.

## B.4 Analyzing the Latent Space

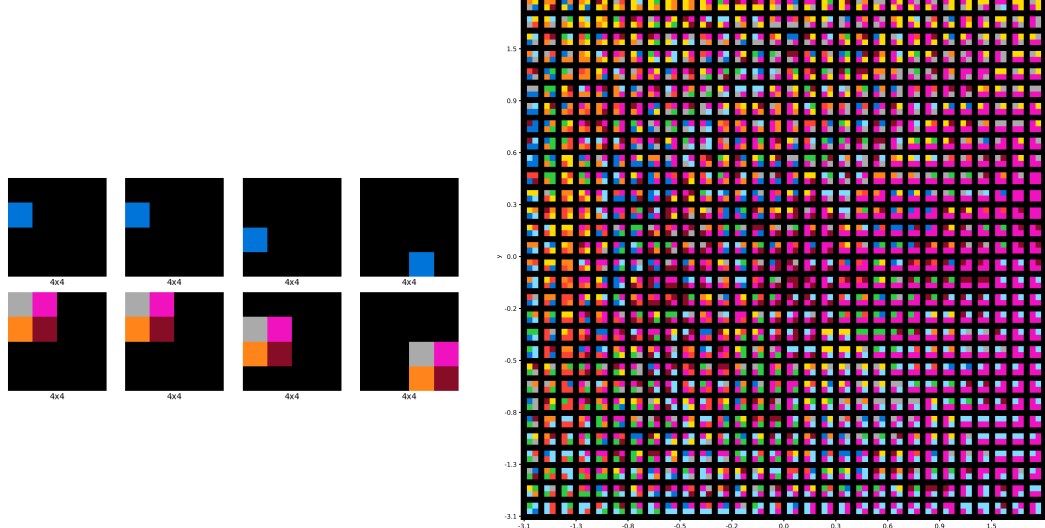

Figure 7: (Left) A 2D pattern task with inputs containing marker points where patterns should be placed, with patterns varying for each program. (Right) The latent traversal visualizes the effect of traversing the latent space, on the predicted pattern by the decoder at marker points.

To validate that the encoder is learning programs in its latent space, we design an even simpler task with small 4x4 grids that have 2x2 patterns. We train a small LPN model until convergence with a latent space of dimension 2 to easily visualize it in figure Figure 7. Due to the simplicity of the task we train the model with *mean* training, i.e. no latent optimization. Because we are using a 2D Gaussian prior for the latent space, we can convert $\mathbb{R}^2$ to the unit square using the normal cumulative distribution function (CDF), and then plot on the unit square at coordinates $(x, y)$ the decoder's output when conditioned by the latent $\text{CDF}(z) = (x, y)$, or equivalently, $z = (\text{PPF}(x), \text{PPF}(y))$ using the percent-point function (PPF). These results demonstrate the diversity and smoothness of the latent space, showing structure in terms of color patterns, which motivates performing gradient ascent latent optimization in more complex tasks. This shows that the latent space can encode a wide range of diversity in its latent space which is especially important for adapting to unseen patterns.

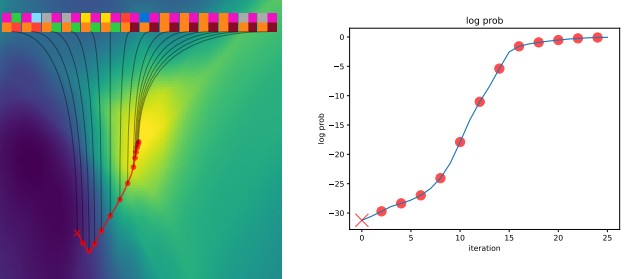

The figures on the left show a random initialization in the latent space and a corresponding latent trajectory following the gradient ascent algorithm. The log-likelihood for each latent is shown during optimization. The decoded pattern is traced until convergence in the latent space, giving insights into how LPN performs latent optimization at test time to find the optimal latent program.

## B.5  Scaling Specification Size

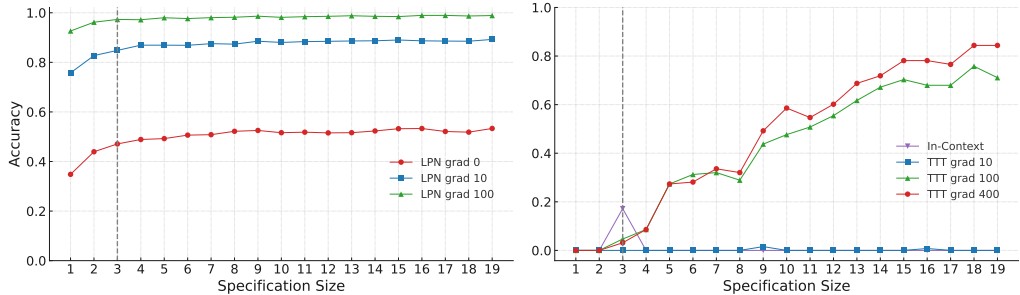

Figure 8: Accuracy of Latent Program Network (LPN) and Test-Time Training (TTT) models as we scale specification size. Higher specification sizes improve accuracy across different training methods. The dashed vertical line at specification size 3 indicates the fixed size used during training.

To analyze the effect of scaling specification size, we conduct an ablation using models trained on the pattern task for a single seed. We compare both in-context learning and LPN. We then evaluate different inference strategies on the in-distribution dataset and analyze their performance as the test specification size increases.

Our results indicate that LPN generalizes effectively across varying specification sizes. This robustness is primarily due to the use of mean pooling, which prevents overfitting to specific context lengths and enables the model to gracefully handle both slightly larger and significantly larger specification sizes. Furthermore, mean pooling encourages representations of the same program to remain structurally similar, improving generalization.

In contrast, in-context learning exhibits strong overfitting to the training specification size, with a noticeable drop in performance as the specification size deviates from the training distribution. However, we observe that fine-tuning-based inference can scale effectively with increasing specification sizes, provided that a sufficient amount of data is available. The effectiveness of fine-tuning is highly dependent on the choice of hyperparameters, requiring substantial amounts of data to maintain strong performance as the specification size grows.

### B.5.1  Performance Across Training Sizes

To further explore the impact of training specification size on model performance, we present results for training sizes 1, 7, and 11, evaluated across test sizes 1, 3, 7, 11, 15, and 19 on the pattern task. The following tables report accuracy for LPN with and without gradient ascent, In-Context Learning, and Test-Time Training (TTT). Notably, In-context learning consistently overfits to specific test sizes that align closely with the training size. In contrast, LPN, particularly with 100 gradient ascent steps, demonstrates robust generalization, maintaining high accuracy across all test sizes and training sizes. TTT shows improved performance as test sizes increase, approaching LPN's accuracy at larger specification sizes when also trained on large specification sizes.

| Method | Test Size | | | | | |
|---|---|---|---|---|---|---|
| | 1 | 3 | 7 | 11 | 15 | 19 |
| LPN grad 0 | 38.9 | 44.1 | 47.5 | 44.7 | 48.8 | 45.9 |
| LPN grad 100 | 89.8 | 92.1 | 94.1 | 94.1 | 95.3 | 96.3 |
| In-Context | 2.1 | 0.0 | 0.0 | 0.0 | 0.0 | 0.0 |
| TTT grad 100 | 7.8 | 19.9 | 50.9 | 72.4 | 81.8 | 90.4 |

Table 7: Performance (top-2 accuracy, percentage) for training size 1 on the pattern task.

| Method | Test Size | | | | | |
|---|---|---|---|---|---|---|
| | 1 | 3 | 7 | 11 | 15 | 19 |
| LPN grad 0 | 14.6 | 27.7 | 32.8 | 33.4 | 33.2 | 35.1 |
| LPN grad 100 | 87.7 | 89.3 | 92.7 | 91.0 | 92.6 | 93.6 |
| In-Context | 0.0 | 0.0 | 78.3 | 0.0 | 0.0 | 0.0 |
| TTT grad 100 | 11.3 | 34.7 | 77.9 | 90.6 | 94.7 | 93.2 |

Table 8: Performance (top-2 accuracy, percentage) for training size 7 on the pattern task.

| Method | Test Size | | | | | |
|---|---|---|---|---|---|---|
| | 1 | 3 | 7 | 11 | 15 | 19 |
| LPN grad 0 | 11.1 | 26.6 | 40.8 | 43.2 | 50.0 | 48.8 |
| LPN grad 100 | 91.4 | 97.1 | 97.5 | 96.9 | 98.0 | 97.9 |
| In-Context | 0.0 | 0.0 | 0.1 | 94.1 | 0.1 | 0.0 |
| TTT grad 100 | 0.1 | 16.2 | 56.1 | 78.3 | 93.7 | 92.2 |

Table 9: Performance (top-2 accuracy, percentage) for training size 11 on the pattern task.

## B.6 Measuring Floating Point Operations

To evaluate the computational efficiency of different methods, we measure the number of floating-point operations (FLOPs) required for performing inference on a single task. The reported FLOP measurements correspond to the cost of generating an entire 30x30 grid given 3 input-output pairs (specification size).

While gradient-based approaches do not require an additional computational budget in terms of trial attempts, their inference mechanisms involve performing gradient updates, either in parameter space or latent space, which adds a computational overhead. A key distinction between TTT and LPN is their computational cost: LPN updates only the latent space by backpropagating through the decoder, whereas TTT requires backpropagation through all model parameters. This makes TTT significantly more expensive and less efficient for real-time applications.

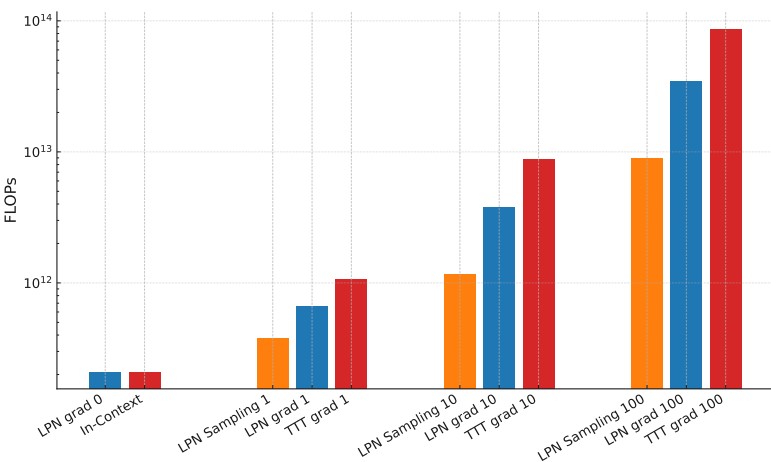

Figure 9: Floating-point operations (FLOPs) required for test-time inference across different methods.

The LPN Sampling results can be understood as ablation of LPN without gradient-based search, but only sampling multiple latents from the encoder computing the decoder likelihood for each, avoiding the computational cost of backpropagating through the decoder. However, this efficiency gain comes at a significant cost in performance, as shown in the pattern task results Table 1.

## B.7 Sequence Ablation

### B.7.1 Dataset

To investigate the generalizability of our results beyond environments with significant spatial structure, we perform an ablation on a synthetic sequence task and replicate the analysis previously conducted on the pattern dataset. This synthetic dataset features a vast program space, with over 100 million unique programs, each defined by composing 3 to 5 parameterized rules that transform sequences of numbers (ranging from 0 to 4). Programs are composed of 3 to 5 rules with specific integer-based parameters (e.g., thresholds or operation values). These rules are then applied in their chosen order: for each rule, we process the sequence from left to right, transforming each position according to the rule's condition and operation, producing an intermediate sequence that serves as input to the next rule. This sequential, rule-by-rule application, with each rule scanning left to right, enables complex transformations and vast numbers of programs with meaningfully different outputs. The core rules are defined as follows:

- If a number is greater than its right neighbor by a threshold $k$, decrease it by a value $m$, modulo 5.
- If a number has identical neighbors on both sides, multiply its value by a factor $n$ and clip the result to 4.
- If a number is greater than its left neighbor by a threshold $k$, add a value $m$, modulo 5.
- If a number is less than its right neighbor by a threshold $k$, subtract a value $m$, modulo 5.
- Replace the number with a function of its neighbors (e.g., sum, average, maximum, or minimum), modulo 5.

These parameterized rules may interact at the same position across their sequential applications, creating cascading effects that result in complex transformations. While a single input-output pair may not uniquely identify the underlying program, multiple pairs typically provide sufficient information to determine it. The table below presents accuracy metrics for models evaluated on this task.

## B.8 LPN fine tuning

We also ablate LPN training by training in full Grad 0 mode for 100k steps. We show the results compared to fine tuning for 5k steps in Grad 1 mode.

| FLOPs | LPN Grad 1 Tune | LPN Grad 0 Tune |
|-------|-----------------|-----------------|
| 2E+11 | 7.75 | 8.25 |
| 2E+12 | 10.25 | 10.25 |
| 2E+13 | 15.25 | 13.60 |
| 2E+14 | 15.50 | 15.10 |
| 2E+15 | 15.50 | 15.10 |

Table 10: Performance of LPN with and without gradient tuning on ARC-AGI for 10k steps

We find a marginal performance gain at higher computational budgets from fine-tuning with one gradient step, so we adopt this approach in the main method. However, we note a slight performance drop in Grad 0 inference, which is expected as the network begins to optimize with gradient adaptation.

**B.9    ARC-AGI Solution Analysis**

In this section we investigate whether there are differences between the types of tasks in ARC-AGI evaluation dataset that LPN solves vs Test-Time Training. We run both methods at the budget of 2E+14 and analyze the problems solved. We first analyze the overlap between problems solved between the two methods. We see that there is a spread of the different problems being solved by the different architectures, a roughly even split between problems only solved by TTT, problems solved only by LPN and problems solved by both.

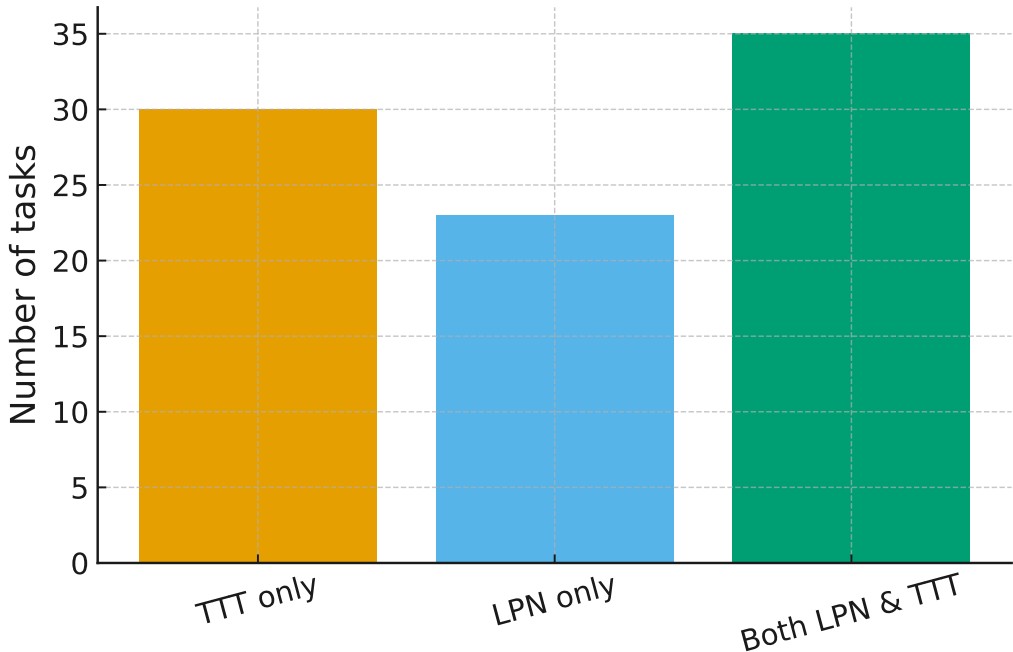

Figure 10: Bar Chart showing Problems Uniquely solved by LPN and TTT and tasks solved when at least one solves the problem

If we measure the accuracy when either LPN or TTT solves an ARC puzzle. We achieve a combined score of 22%. We also show 3 examples of problems solved only by LPN and 3 examples only solved by Test-Time Training to understand the differences between the problems each method solved.

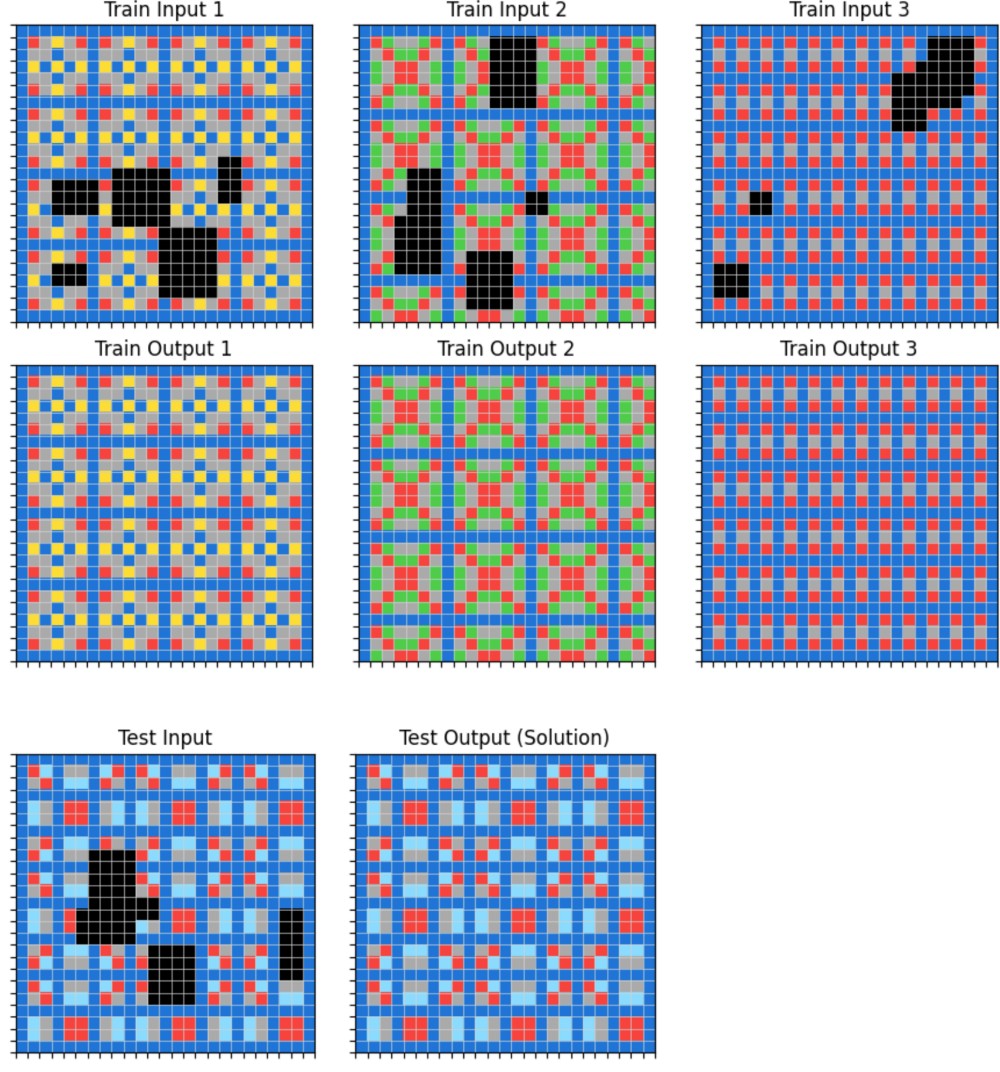

Figure 11: LPN Example 1

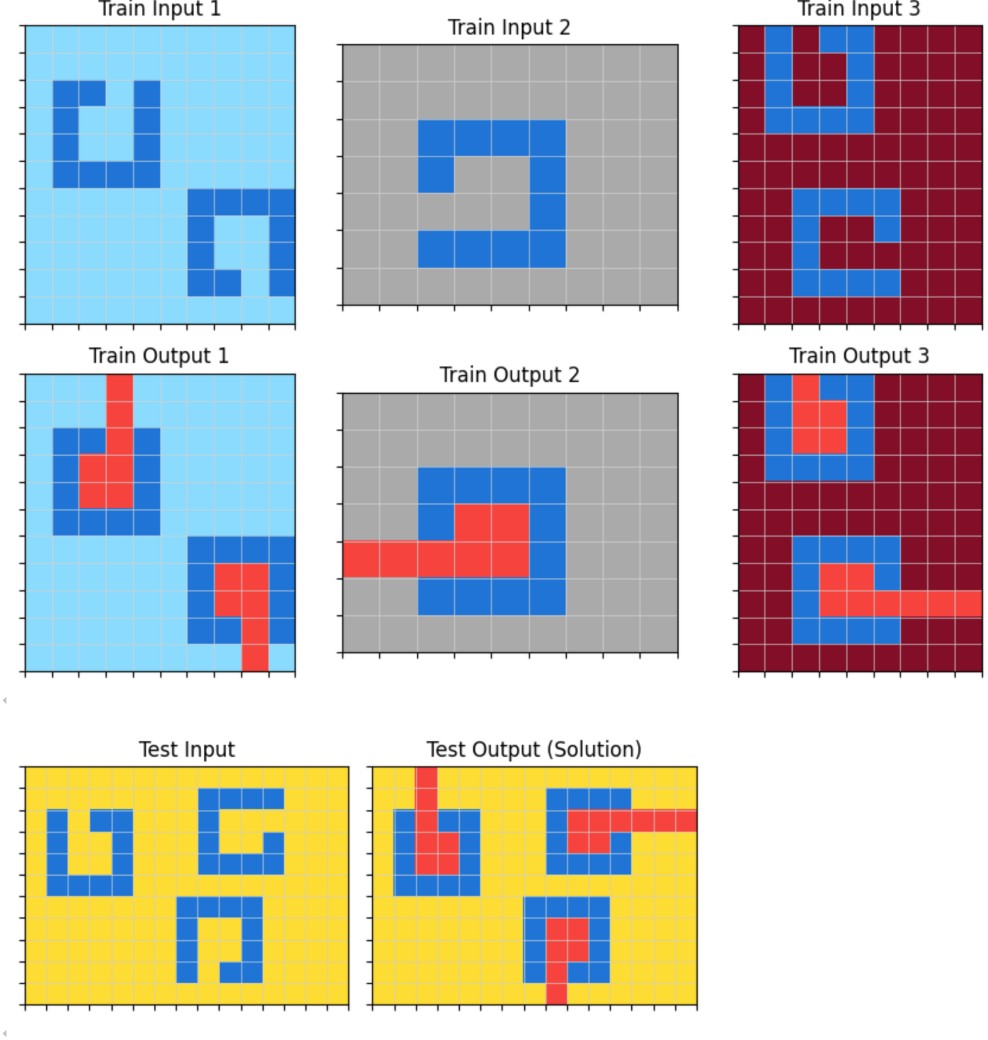

Figure 12: LPN Example 2

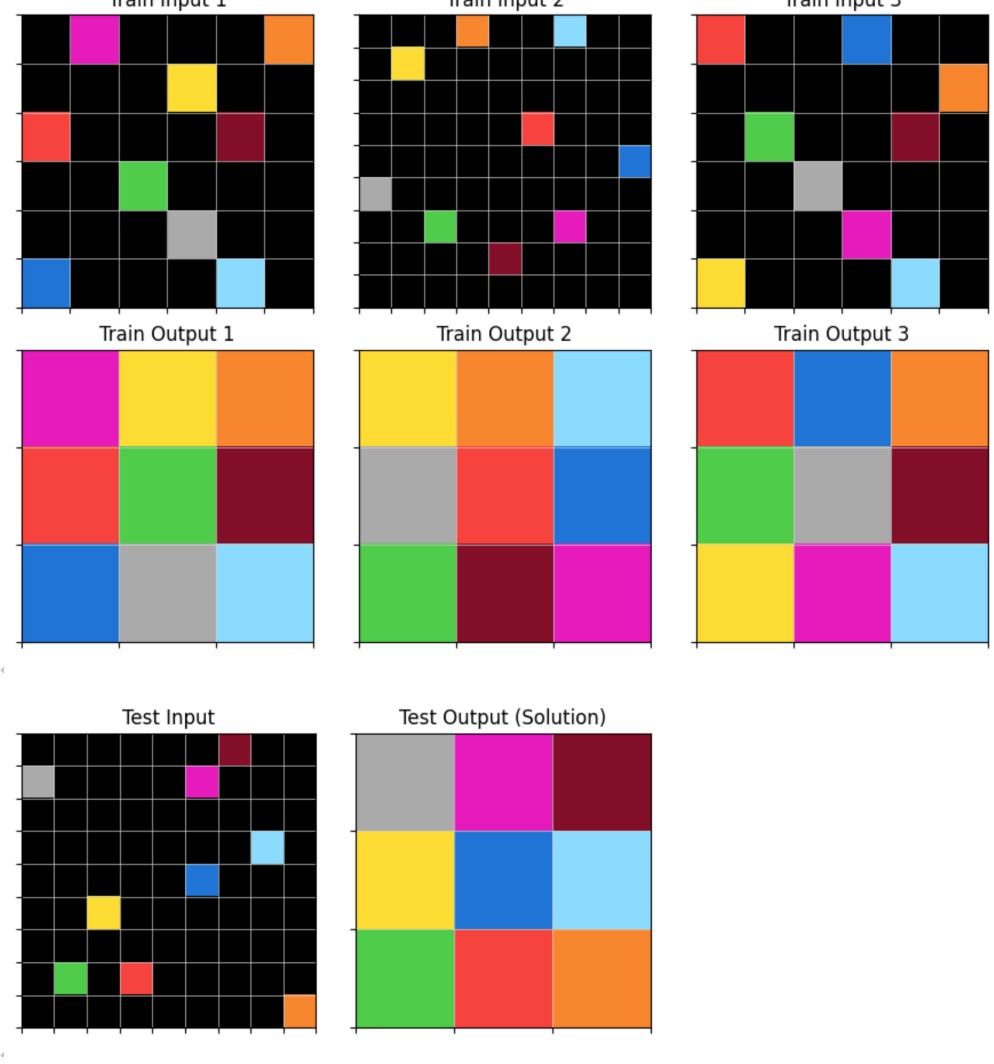

Figure 13: LPN Example 3

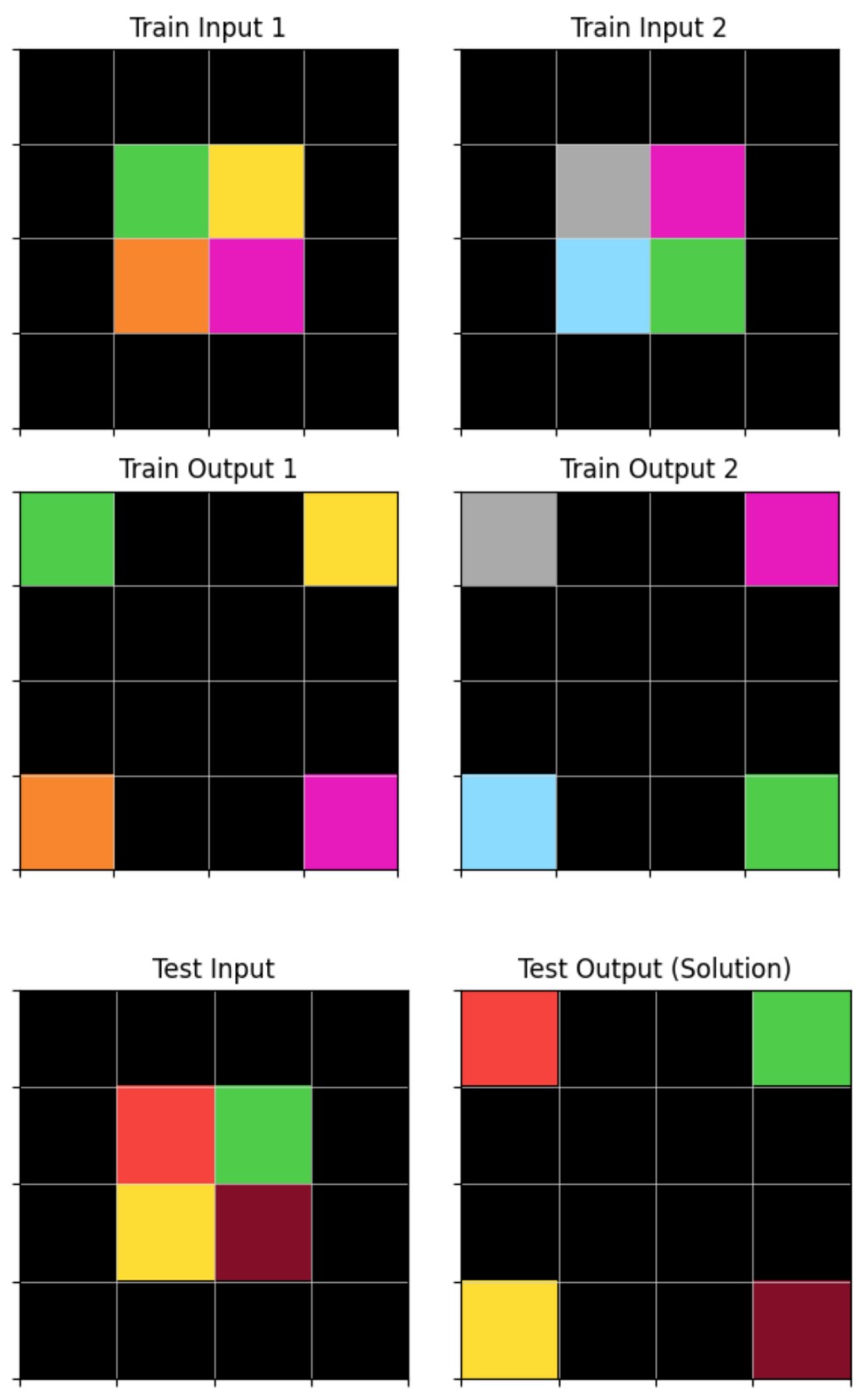

Figure 14: TTT Example 1

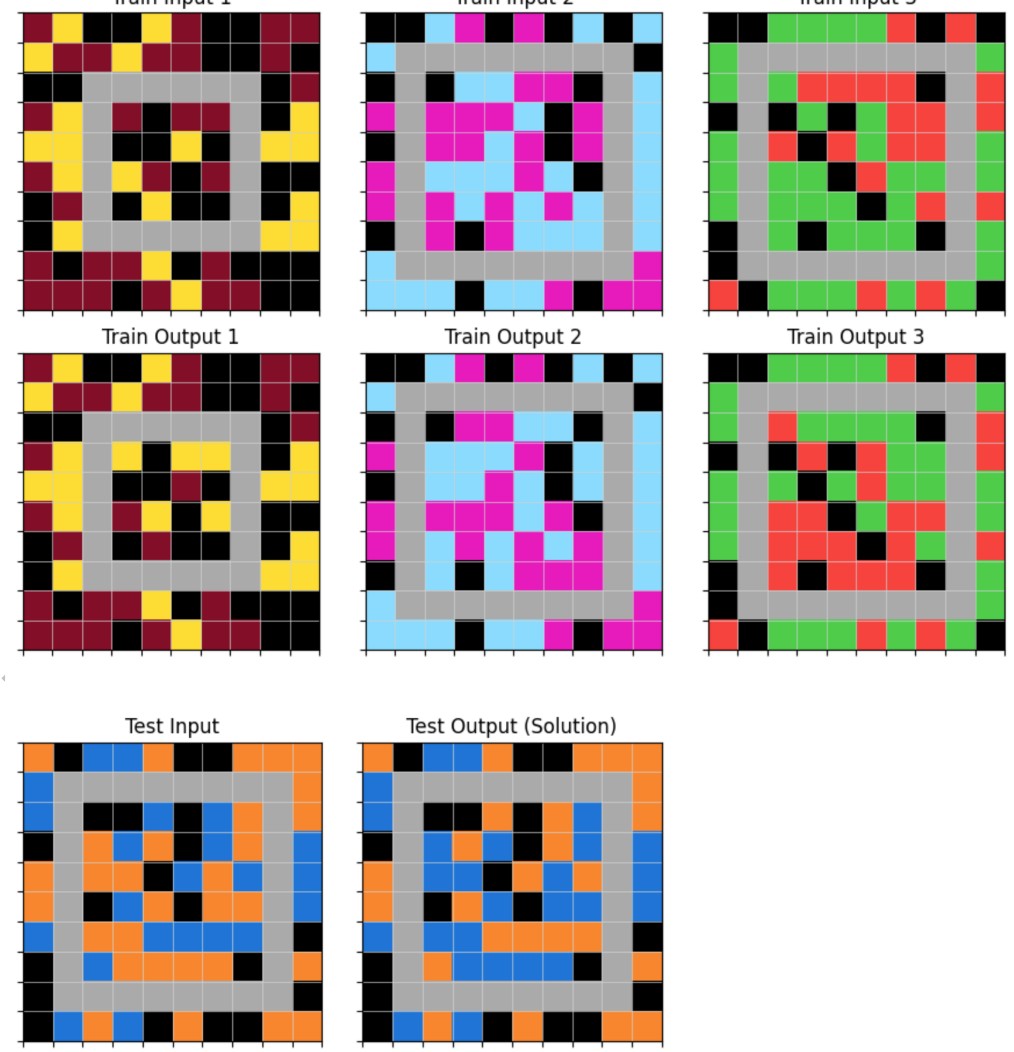

Figure 15: TTT Example 2

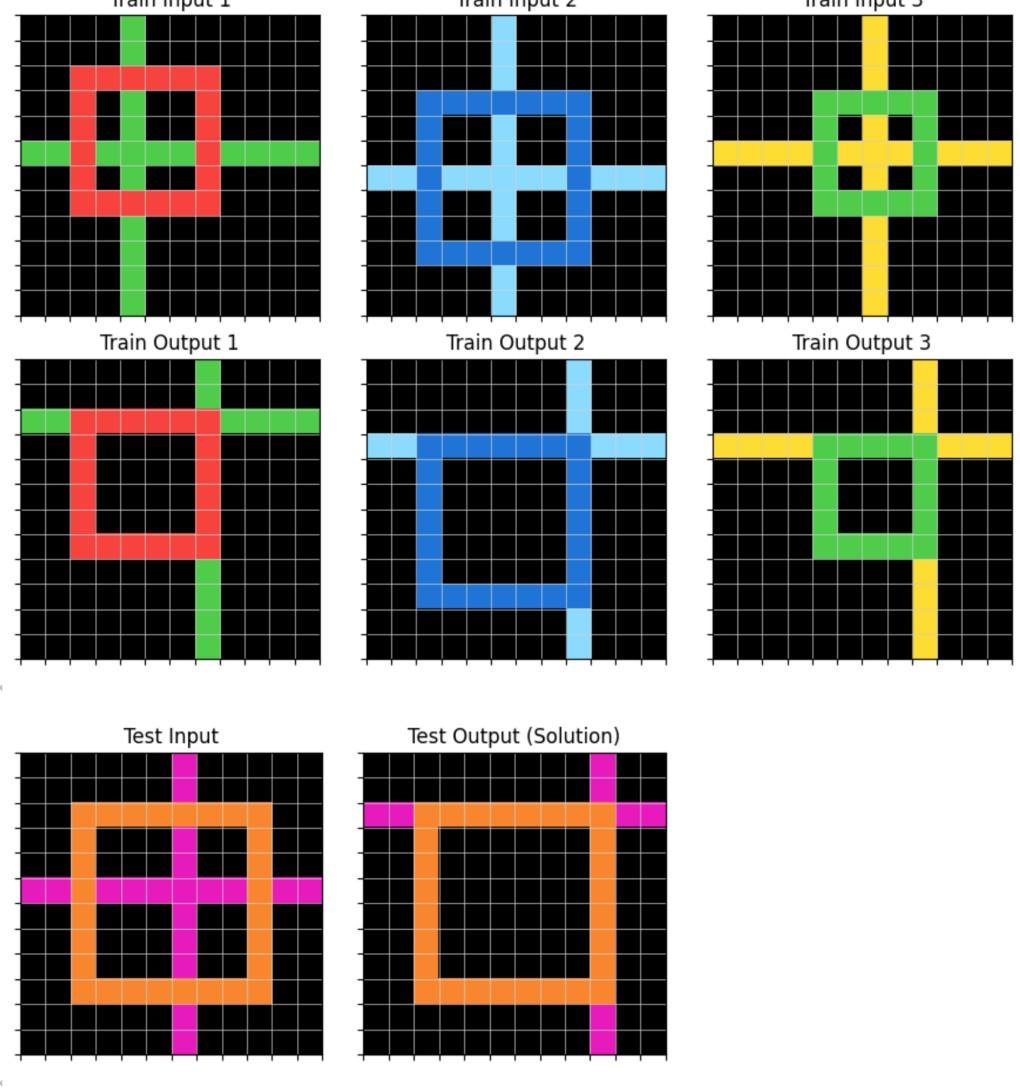

Figure 16: TTT Example 3

## B.10  Investigating Composition

To evaluate the generalization capabilities of LPN to novel compositions of learned primitives, we construct simple test cases by combining two distinct operations from the training set that the model has only encountered separately during training. This setup probes whether LPN can flexibly integrate these primitives to produce the desired composite effect, a key indicator of its ability to handle systematic generalization beyond rote memorization.

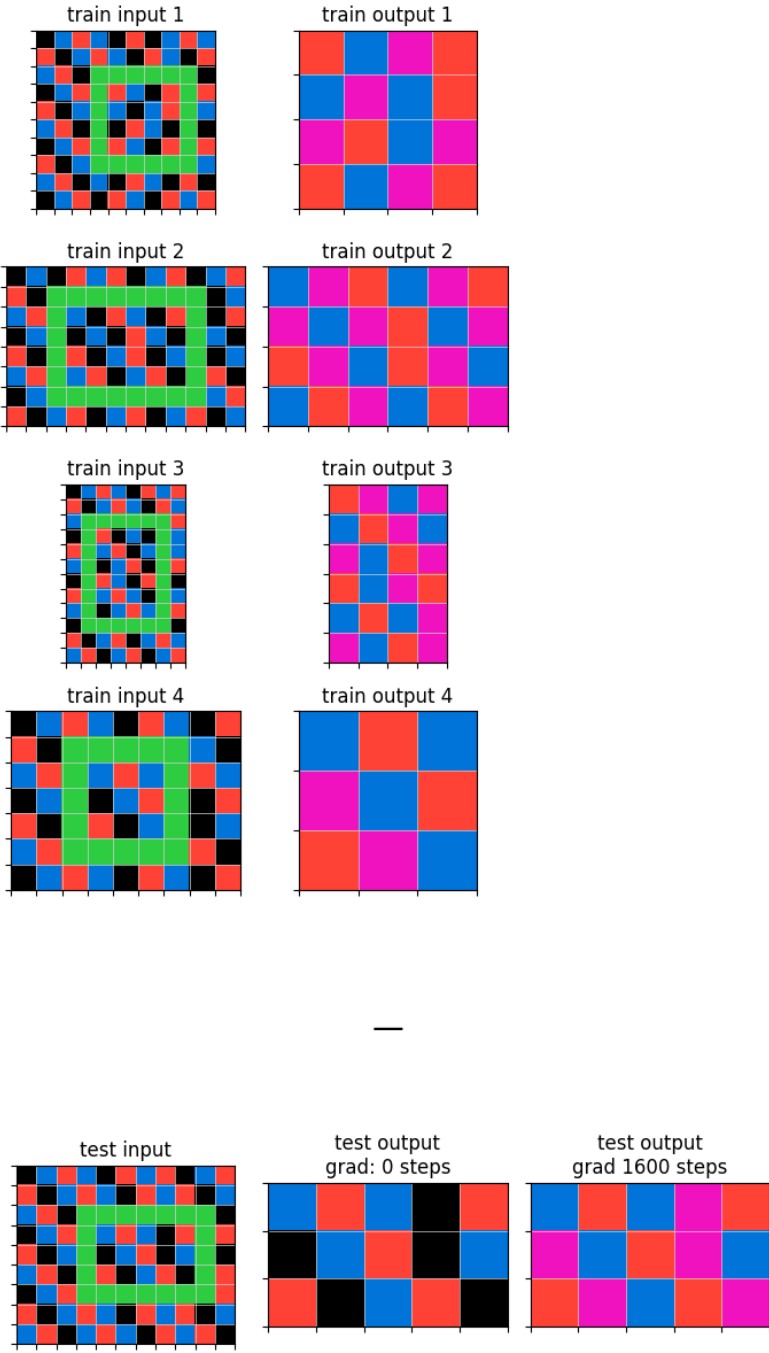

Figure 17: Testing LPN on the composition of two operations previously only seen separately.

In Figure 17, we examine a specific instance where the first operation extracts the pattern within a bounding box, while the second replaces all black pixels with pink. The direct output of the

encoder reveals overfitting to isolated training patterns: it correctly extracts and the pattern within the bounding box, but fails to convert black pixels to black. However, by performing gradient ascent in the latent space to refine the encoding, we can mitigate these errors. This optimization aligns the latent representation more closely with the composite objective, enabling the decoder to accurately render the full pink conversion across the extracted pattern. This demonstrates LPN's potential for compositional reasoning, where latent-space adjustments unlock emergent behaviors not explicitly trained for.

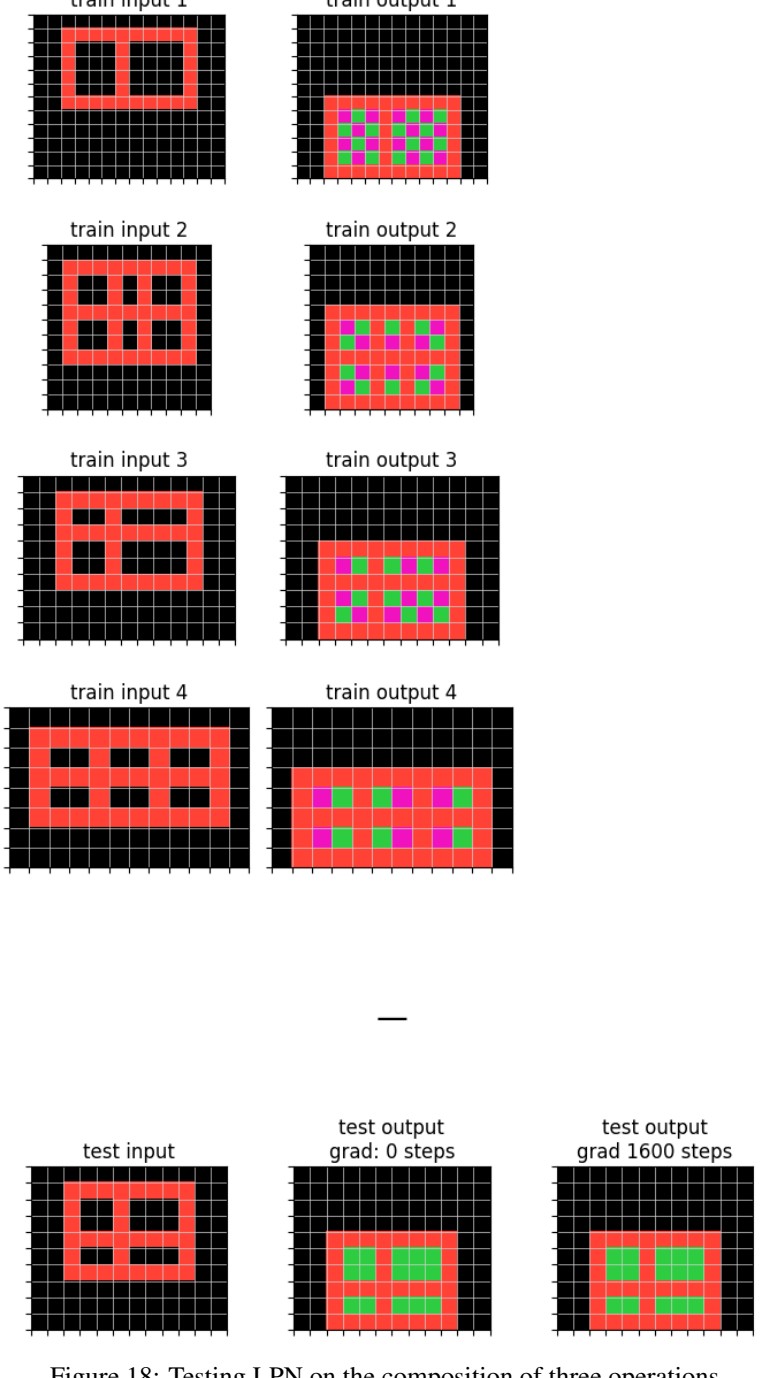

Figure 18: Testing LPN on the composition of three operations.

In Figure 18, we examine the combination of three primitives: a gravity-type operation, a fill operation, and the use of a checkerboard pattern for the fill. While LPN can directly predict the

gravity and fill components, it overfits to the fill operations seen during training. As a result, even with a gradient-based search, it is not able to resolve this error and apply the checkerboard pattern instead.

### B.11 LPN ARC-AGI Inference Ablations

To dissect the contributions of key components to LPN's performance on ARC-AGI, we performed ablations on various test-time inference strategies. These experiments isolate the roles of the program prior and gradient-based search, using the ARC-AGI evaluation set (out-of-distribution relative to Re-ARC). We fix the number of sampled latents during inference and vary the search budgets across $\{10, 50, 100, 200, 400\}$ steps, evaluating on a single seed.

We compare the following inference strategies:

- **Encoder Sampling.** Samples multiple latents from the encoder distribution, evaluates the specification likelihood for each, and selects the highest-scoring one. This assesses structured gradient-based search against brute-force sampling near the encoder output.

- **Gaussian Init + Gradient Search.** Employs the core LPN gradient-based search but initializes from a sample drawn from the Gaussian prior. This isolates the encoder's contributions in providing strong initializations and enhancing overall performance.

- **Encoder Init + Local Search.** Initializes from the encoder output but substitutes gradient updates with a mutation operator (continuous noise perturbations). This evaluates gradient-based optimization against unstructured local search, while permitting exploration beyond the encoder distribution.

- **Gaussian Init + Local Search.** Pairs prior-based initialization with local search. This baseline quantifies the joint necessity of encoder-guided initialization and gradient direction.

- **Encoder Init + Gradient Search (LPN).** The full LPN inference procedure.

Table 11 reports the performance (in %) across these strategies and budgets.

Table 11: Ablation results (%) on ARC-AGI evaluation for inference strategies under varying search budgets.

| Inference Method | 10 | 50 | 100 | 200 | 400 |
|---|---|---|---|---|---|
| Encoder Sampling | 9.12% | 10.75% | 10.13% | 10.50% | 10.62% |
| Gaussian Init + Gradient Search | 1.38% | 2.75% | 5.37% | 6.88% | 10.75% |
| Encoder Init + Local Search | 8.38% | 9.50% | 10.37% | 10.75% | 10.75% |
| Gaussian Init + Local Search | 0.25% | 0.25% | 0.25% | 0.75% | 0.25% |
| **Encoder Init + Gradient Search (LPN)** | **9.10%** | **13.25%** | **15.00%** | **15.50%** | **15.50%** |

These results reveal that repeated encoder sampling yields only marginal gains over a single sample (7.75%), underscoring the limitations of brute-force approaches. The gap between Gaussian Init + Gradient Search and Encoder Init + Gradient Search emphasizes the encoder's dual role: seeding effective starting points for low budgets and amplifying gains at high budgets (e.g., +4.75% at 400 steps). Encoder Init + Local Search matches gradient search at low budgets (10 steps) but plateaus thereafter, highlighting gradient guidance's superiority for deeper exploration. Gaussian Init + Local Search yields near-zero performance, confirming the indispensability of encoder initialization paired with directed optimization. In aggregate, Encoder Init + Gradient Search (LPN) dominates all baselines, validating the encoder-informed initialization and gradient-based refinement as pivotal to LPN's efficacy on ARC-AGI.

## C   LPN Algorithm

Below we outline two algorithms: first, LPN test-time inference (Algorithm 1) and its mechanism for performing inductive inference. Second, we provide the full algorithm for LPN during training (Algorithm 2).

---

**Algorithm 1** LPN Test-Time Inference with Gradient Ascent Latent Optimization

---

**Require:** $n$ input-output pairs $(x_i, y_i)$, a test input $x_{n+1}$, number of gradient steps $K$
  1: **for** $i = 1, \ldots, n$ **do**                                                    ▷ Can be done in parallel
  2:     Sample $z_i \sim q_\phi(z|x_i, y_i)$
  3: **end for**
  4: Initialize latent $z' \leftarrow \frac{1}{n} \sum_{i=1}^n z_i$
  5: **for** $k = 1, \ldots, K$ **do**                                                  ▷ Perform gradient ascent
  6:     $z' \leftarrow z' + \alpha \cdot \nabla_z \sum_{i=1}^n \log p_\theta(y_i|x_i, z)|_{z=z'}$
  7: **end for**
  8: **return** $y_{n+1} \sim p_\theta(y|x_{n+1}, z')$

---

**Algorithm 2** LPN Training with Gradient Ascent Latent Optimization

---

**Require:** Encoder parameters $\phi$, decoder parameters $\theta$
  1: **for** $t = 1, \ldots,$ num_training_steps **do**
  2:     Sample $n$ input-output pairs $(x_i, y_i)$ from the same program
  3:     **for** $i = 1, \ldots, n$ **do**                                     ▷ Can be done in parallel
  4:         Sample $z_i \sim q_\phi(z|x_i, y_i)$                ▷ Using the reparameterization trick
  5:     **end for**
  6:     **for** $i = 1, \ldots, n$ **do**                                     ▷ Can be done in parallel
  7:         $z_i' \leftarrow \frac{1}{n-1} \sum_{\substack{j=1 \\ j \neq i}}^n z_j$
  8:         **for** $k = 1, \ldots, K$ **do**                    ▷ Perform gradient ascent in the latent space
  9:            $z_i' \leftarrow z_i' + \alpha \cdot \nabla_z \sum_{\substack{j=1 \\ j \neq i}}^n \log p_\theta(y_j|x_j, z)|_{z=z_i'}$  ▷ Optional stop-grad on the 2nd term
10:         **end for**
11:         $\mathcal{L}_i \leftarrow -\log p_\theta(y_i|x_i, z_i') + \beta \cdot D_{\text{KL}}(q_\phi(z|x_i, y_i) \| \mathcal{N}(0, \text{I}))$
12:     **end for**
13:     $\mathcal{L} \leftarrow \frac{1}{n} \sum_{i=1}^n \mathcal{L}_i$                                          ▷ Total loss for all pairs
14:     Update $\phi$ and $\theta$ via gradient descent on $\mathcal{L}$
15: **end for**

---

# D   Variational Inference

Latent Program Networks (LPNs), when operated in grad 0 mode, perform a form of variational inference. Updating the latent representation using gradients from the encoder constitutes semi-amortized variational inference [Kim et al., 2018, Marino et al., 2018].

In prior work, such as LEAPs Trivedi et al. [2021], it is assumed during training that the full representation of a program $f$ is observable. In this setting, variational inference on programs can be conducted using a standard Variational Autoencoder (VAE) restricted to the program space. LEAPs introduce a form of variational inference via an Execution Evidence Lower Bound (ELBO), where function reconstruction is based on correctly executing the function for a given input rather than reconstructing the full program representation. The Execution ELBO is defined as:

$$\mathbb{E}_{z \sim q_\phi(z|f)} \left[ \mathbb{E}_{x \sim p(x)} \left[ \log p_\theta(y = f(x) \mid x, z) \right] \right] - \text{KL}(q_\phi(z|f) \parallel p(z)).$$

This formulation is advantageous because it enables learning a function that directly executes the program without requiring explicit reconstruction of the program followed by execution. In LPNs, this is critical, as the differentiable parameterization of the function executor allows backpropagation through the executor to perform program search in the latent space.

In this work, we assume that the functions generating the underlying input-output pairs are not fully observable, aligning with real-world scenarios where the data-generating functions are rarely fully known. Instead, we observe only partial data for each function, represented as a dataset $X_d = \{(x_{i,d}, y_{i,d})\}_{i=1}^{N_d}$, where each dataset $d$ corresponds to a set of input-output pairs generated by a particular unseen function. Across $D$ such datasets, the following objective serves as a practical approximation to the Execution ELBO:

$$\sum_{d=1}^{D} \sum_{i=1}^{N_d} \mathbb{E}_{z \sim q_\phi(z|X_d)} \left[ \log p_\theta(y_{i,d} \mid x_{i,d}, z) \right] - \text{KL}(q_\phi(z|X_d) \parallel p(z)), \tag{7}$$

where $D$ is the number of datasets, each representing input-output pairs from a distinct unseen function, and $N_d$ is the number of samples in dataset $d$. However, this standard objective is flawed because the encoder $q_\phi(z|X_d)$ can "cheat" by encoding specific details of the pair $(x_{i,d}, y_{i,d})$ it needs to reconstruct, leading to memorization rather than capturing the general function $f$. To address this, we propose the Leave-One-Out (LOO) objective:

$$\sum_{d=1}^{D} \sum_{i=1}^{N_d} \mathbb{E}_{z \sim q_\phi(z|X_{-i,d})} \left[ \log p_\theta(y_{i,d} \mid x_{i,d}, z) \right] - \text{KL}(q_\phi(z|X_d) \parallel p(z)), \tag{8}$$

where $X_{-i,d} = X_d \setminus \{(x_{i,d}, y_{i,d})\}$ denotes the dataset $X_d$ excluding the $i$-th input-output pair. This objective prevents memorization by denying the encoder access to $(x_{i,d}, y_{i,d})$ when producing the latent representation $z$ used for its reconstruction. Consequently, $q_\phi$ must infer the underlying function from the remaining data $X_{-i,d}$, making the LOO objective a better functional approximation of the Execution ELBO's intent by directly promoting generalization. Training optimizes the encoder $q_\phi(z|X_d)$ and decoder $p_\theta(y \mid x, z)$ across multiple datasets, resulting in an amortized inference model $q_\phi$ that efficiently proposes an approximate posterior for any given dataset $X_d$.

For optimal performance on a specific test dataset $X_{\text{test}}$, we employ semi-amortized variational inference. First, the trained amortized encoder $q_\phi(z|X_{\text{test}})$ rapidly generates an initial latent representation $z_0$. Then, with the encoder parameters $\phi$ and decoder parameters $\theta$ fixed, we perform instance-specific optimization starting from $z_0$ to find a refined latent representation $z^*$. This is achieved by directly maximizing the ELBO via latent $z$. This combination of an efficient amortized proposal and instance-specific ELBO refinement yields a superior latent program representation $z^*$ tailored to the test problem. This procedure, known as semi-amortized variational inference, balances computational efficiency with high-quality inference.

# E  Hyperparameters

In this section, we outline our approach to hyperparameter search and provide full documentation of all the hyperparameters used in all reported experiments.

**Hyperparameter Search**  To ensure a fair comparison between LPN and in-context learning, which share the same core architecture for embedding inputs and generating outputs, we kept all architectural parameters identical across methods. We conducted hyperparameter testing to determine whether to use rotational embeddings. We performed testing by repeating the decoder validation experiment with and without rotational embeddings. Rotational embeddings improved performance across all individual tasks and so was selected to be used in both the encoder and decoder of LPN.

For the ARC-AGI results, we performed a hyperparameter search over the learning rate for test time adaptation for both test-time tuning and LPN. Since these methods operate in different spaces, they are unlikely to require similar parameters, making it fairer to tune this parameter independently for each baseline. We validated on a held-out test set of unseen problems from the RE-ARC dataset. We searched over learning rates ranging from $10^{-1}$ to $10^{-7}$. Performance was evaluated by measuring average accuracy across five FLOP measures (ranging from 2E+11 to 2E+15). For the baseline pattern task in Table 1 we choose a learning rate of 0.1 without performing hyperparameter optimisation as the experiment is simply to understand the behavior of LPN and not to optimize performance. For pattern OOD task we perform grid search for LPN and TTT over a dataset of from the strongly OOD task for 10 gradient steps, filtering out learning rates that do not decrease the test-time loss by at least 1.0%. We use these learning rates for the scaling specification size ablation also.

**Validating the Decoder**  In section B.3, we train an LPN model on each of the `re-arc` generators corresponding to the first 5 tasks from the training set. For each task, we train the model for 10k gradient steps with a batch size of 128 and 4 pairs per specification, resulting in 5,120,000 input-output pairs. We gather all hyperparameters in table 12, common to all the tasks except `045e512c` which had 4 encoder layers, 8 heads, 32 embedding dimensions per head, an MLP factor of 2.0, for a total of 8.7M parameters.

| Component | Hyperparameter | Value |
|---|---|---|
| Encoder Transformer | Number of Layers | 0 |
| | Number of Heads | 6 |
| | Embedding Dimension per Head | 16 |
| | Latent Dimension | 32 |
| | RoPE | False |
| Decoder Transformer | Number of Layers | 3 |
| | Number of Heads | 6 |
| | Embedding Dimension per Head | 16 |
| | MLP Dimension Factor | 1.0 |
| | RoPE | False |
| Training | Number of Parameters | 829k |
| | Training Steps | 10k |
| | Batch Size | 128 |
| | Optimizer | AdamW |
| | Gradient Clipping Norm | 1.0 |
| | Learning Rate | 4e-4 |
| | Number of Rows & Columns | 30, 30 |

Table 12: Hyperparameters for the experiments from section B.3, validating the decoder.

**Pattern Task** In section 5.2, we train an LPN model on a 10x10 task with 4x4 patterns. We train each method (mean, gradient ascent, etc) for a total of 20k steps with a batch size of 128 and 4 pairs per specification, resulting in a total of 10M input-output pairs.

| Component | Hyperparameter | Value |
|---|---|---|
| Encoder Transformer | Number of Layers | 2 |
| | Number of Heads | 6 |
| | Embedding Dimension per Head | 16 |
| | MLP Dimension Factor | 1.0 |
| | Latent Dimension | 32 |
| | RoPE | False |
| Decoder Transformer | Number of Layers | 2 |
| | Number of Heads | 6 |
| | Embedding Dimension per Head | 16 |
| | MLP Dimension Factor | 1.0 |
| | RoPE | False |
| Training | Number of Parameters | 973k |
| | Training Steps | 20k |
| | Batch Size | 128 |
| | Prior KL Coeff | 1e-4 |
| | Optimizer | AdamW |
| | Gradient Clipping Norm | 1.0 |
| | Learning Rate | 4e-4 |
| | Number of Rows & Columns | 10, 10 |
| Testing | TTT Learning Rate | 1e-4 |
| | LPN Learning Rate | 1e-1 |
| | Optimizer | Adam |

Table 13: Hyperparameters for the experiments from section 5.2, i.e. the pattern task.

**Analyzing the Latent Space** In section B.4, we train a small LPN model on a reduced version of the *Pattern* task with grids of size 4x4 and patterns of size 2x2. We used the following hyperparameters for training in table 14.

| Component | Hyperparameter | Value |
|---|---|---|
| Encoder Transformer | Number of Layers | 2 |
| | Number of Heads | 6 |
| | Embedding Dimension per Head | 12 |
| | MLP Dimension Factor | 4.0 |
| | Latent Dimension | 2 |
| | RoPE | False |
| Decoder Transformer | Number of Layers | 2 |
| | Number of Heads | 6 |
| | Embedding Dimension per Head | 12 |
| | MLP Dimension Factor | 4.0 |
| | RoPE | False |
| Training | Number of Parameters | 1M |
| | Training Steps | 200k |
| | Batch Size | 128 |
| | Prior KL Coeff | 1e-3 |
| | Optimizer | AdamW |
| | Gradient Clipping Norm | 1.0 |
| | Learning Rate | 4e-4 |
| | Number of Rows & Columns | 4, 4 |
| Testing | TTT Learning Rate | 1e-4 |
| | LPN Learning Rate | 1e-1 |
| | Optimizer | Adam |

Table 14: Hyperparameters for the experiment in section B.4, i.e. analyzing the latent space.

**Out-Of-Distribution**    In section 5.5, we train LPN models similar to those above in the *Pattern task* and evaluate them on different distributions. We gather hyperparameters used for training in table 15.

| Component | Hyperparameter | Value |
|---|---|---|
| Encoder Transformer | Number of Layers | 4 |
| | Number of Heads | 8 |
| | Embedding Dimension per Head | 8 |
| | MLP Dimension Factor | 2.0 |
| | Latent Dimension | 32 |
| | RoPE | False |
| Decoder Transformer | Number of Layers | 2 |
| | Number of Heads | 8 |
| | Embedding Dimension per Head | 4 |
| | MLP Dimension Factor | 1.0 |
| | RoPE | False |
| Training | Number of Parameters | 1M |
| | Training Steps | 100k |
| | Batch Size | 128 |
| | Prior KL Coeff | 1e-4 |
| | Optimizer | AdamW |
| | Gradient Clipping Norm | 1.0 |
| | Learning Rate | 4e-4 |
| | Number of Rows & Columns | 10, 10 |
| Testing | TTT Learning Rate | 1e-5 |
| | LPN Learning Rate | 1e-1 |
| | Optimizer | Adam |

Table 15: Hyperparameters for the experiments from section 5.5, i.e. the study of out-of-distribution performance of LPN on the Pattern task.

**ARC-AGI**  In table 16, we finally present the hyperparameters used for experiments on ARC-AGI (section 5.7).

| Component | Hyperparameter | Value |
|---|---|---|
| Encoder Transformer | Number of Layers | 8 |
| | Number of Heads | 8 |
| | Embedding Dimension per Head | 64 |
| | MLP Dimension Factor | 4.0 |
| | Latent Dimension | 256 |
| | RoPE | True |
| | RoPE max freq | 10 |
| Decoder Transformer | Number of Layers | 6 |
| | Number of Heads | 8 |
| | Embedding Dimension per Head | 64 |
| | MLP Dimension Factor | 4.0 |
| | RoPE | True |
| | RoPE max freq | 10 |
| Training | Number of Parameters | 178M |
| | Training Steps | 100k |
| | Batch Size | 256 |
| | Prior KL Coeff | 1e-4 |
| | Optimizer | AdamW |
| | Gradient Clipping Norm | 1.0 |
| | Learning Rate | 3e-4 |
| | Number of Rows & Columns | 30, 30 |
| Testing | TTT Learning Rate | 1e-4 |
| | LPN Learning Rate | 5e-2 |
| | Optimizer | Adam |

Table 16: Hyperparameters for the experiment in section 5.7, i.e. training LPN to solve the ARC-AGI benchmark.

# F  Additional Charts

## F.1  Latent Program Embeddings

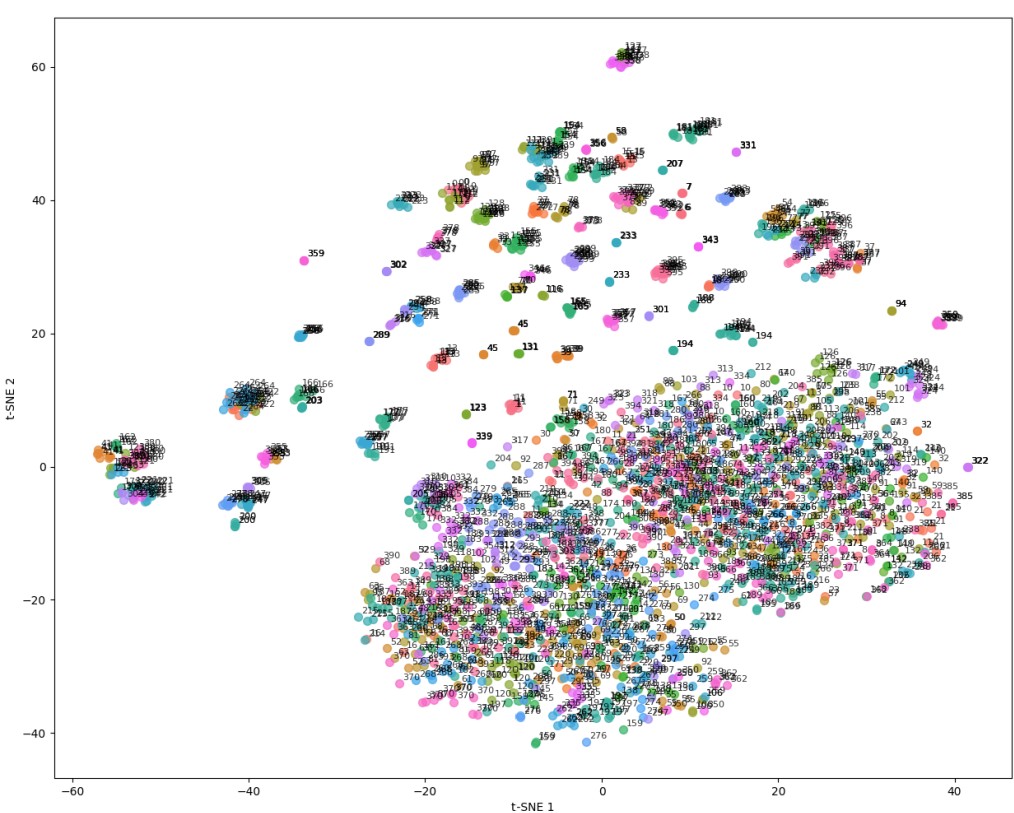

Figure 19: T-SNE visualization of the latent space of input-output pairs sampled from the `re-arc` generators. We see strong evidence that the latent embeddings encode information about programs with significant clustering in the latent space for the same programs across different input output pairs.

## F.2 Decoder Gradient Field

Figure 20 visualizes the likelihood landscape of decoding the correct output conditioned on the given input for a single pattern task, while varying the latent input. The gradient contours overlaid on the plot illustrate the optimization dynamics when performing gradient-based updates in this space. The trajectories depict how gradient ascent seeks to maximize the decoding likelihood, revealing the structure of the landscape. Certain regions form basins of attraction that lead to valid solutions, while others correspond to local optima where the likelihood of decoding the correct output stagnates. This visualization highlights both the feasibility of optimizing the latent representation and the potential challenges of escaping suboptimal regions in the latent space.

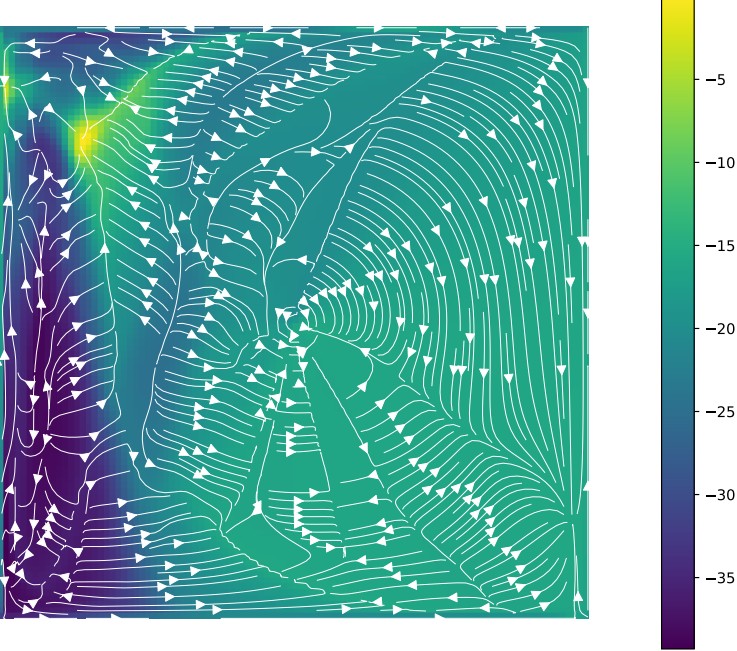

Figure 20: Visualization of the likelihood landscape for decoding the correct output conditioned on the input, as a function of the latent space.

# G   Architecture

In all our experiments, programs are defined in the input-output space of ARC-AGI, i.e., 2D grids whose cells can take 10 different values and have shapes $(n, m)$ with $n, m \in [1, 30]$. We implement both the encoder and decoder as small transformers [Vaswani et al., 2017] specifically designed for this benchmark, in contrast to the more general large language models (LLMs) typically used [Wang et al., 2023].

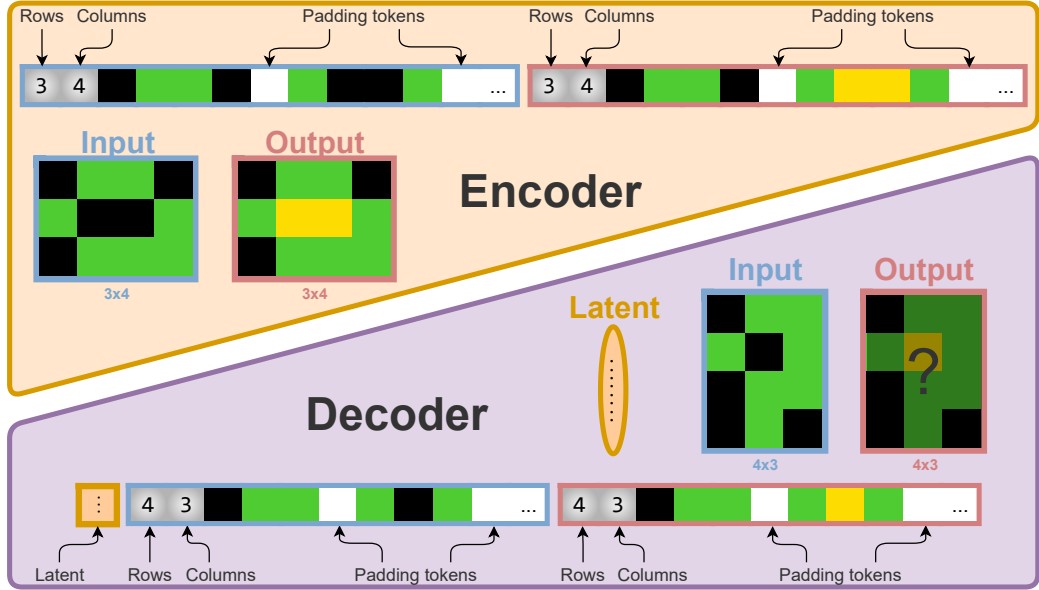

Figure 21: LPN architecture for ARC-AGI. Both the encoder and decoder are small transformers that take flattened padded grids as inputs. The actual number of rows and columns is prefixed to each sequence.

We model the input and output images as 2D grids, which we pad and flatten in a raster-scan fashion to form sequences of pixel values, each of size $30 \times 30 = 900$ (see Figure 21). Each grid sequence is prefixed with shape information, namely two extra values for the number of rows and columns, resulting in sequences of 902 values. For both the Encoder and Decoder grid positions are encoded using RoPE [Su et al., 2024] for both row and column indices.

## G.1   Encoder

The encoder processes both the input and output grids from a given pair and returns a distribution of inferred program latents underlying the task. Specifically, it outputs the mean and diagonal covariance of a multivariate normal distribution from which latents can be sampled. Each grid sequence contains 902 values, and we add an extra CLS token for the output embedding, resulting in a total sequence length of 1805 for the encoder transformer. The encoder is implemented as a standard transformer [Vaswani et al., 2017] with pre-layer normalization [Baevski and Auli, 2018, Xiong et al., 2020] and multi-head attention. To incorporate identify the input from the output we add an embedding emb$(c)$, $c \in \{0, 1\}$ is the channel index (0 for input, 1 for output). All 1800 color values (0 to 9), four shape values (1 to 30), and the CLS token are separately embedded into $\mathbb{R}^H$ using lookup tables. Padded tokens, determined by the shape values, are masked, and the sequence is fed into multiple transformer blocks, see Section E for hyperparameter details. In the encoder the attention mask is non-causal, allowing all non-padded tokens to attend to each other during encoding. The CLS embedding is passed through a layer normalization and two parallel dense layers to output the mean and diagonal log-covariance of the multivariate normal distribution over latents. Sampled program latents have dimension $d$, which may differ from the embedding dimension $H$.

---

https://github.com/crowsonkb/rope-flax

## G.2 Decoder

The decoder takes an input grid and a latent program and autoregressively generates an output grid. Its design is similar to the encoder, with key differences. First, the flattened sequence is prefixed with a projection of the latent embedding. Since the decoder generates the output autoregressively, the attention mask is causal on the output grid portion of the sequence (the second half). The attention mask also dynamically accounts for padding tokens based on the predicted output shapes. The sequence embeddings corresponding to the output are extracted and projected to either shape logits for the first two embeddings or grid logits for the 900 output grid embeddings. Each output token embedding maps to logits for the next token in a raster-scan fashion. However, due to padding at each row, the last embedding of each row is mapped to the first token of the next row.

## H Baselines

### H.1 Transductive Baseline

We compare LPN against a transductive baseline that directly conditions on the specification without explicitly constructing a latent program representation. This approach, similar to Kolev et al. [2020], Li et al. [2024a], processes the specification by encoding and concatenating each input-output pair separately.

**Encoder:** The encoder maps each input-output pair to an encoding vector:

$$z_i = e_\phi(x_i, y_i) \quad \forall i \in [1, n] \tag{9}$$

where $e_\phi$ is a neural network parameterized by $\phi$ that processes individual input-output pairs.

**Concatenation:** Unlike LPN which searches for a single latent program, the transductive baseline uses each encoding as a token embedding in the input sequence to the transformer. Specifically, the encodings are concatenated as:

$$z_{\text{cat}} = [z_1; z_2; \ldots; z_n] \tag{10}$$

where $[;]$ denotes sequence concatenation and each $z_i$ serves as a token embedding in the transformer's input sequence.

**Decoder:** The transformer decoder processes this sequence of embeddings along with the new input to generate the output:

$$\hat{y}_{n+1} \sim p_\theta(y|x_{n+1}, z_{\text{cat}}) \tag{11}$$

where the decoder attends to both the new input $x_{n+1}$ and the sequence of specification embeddings $z_{\text{cat}}$.

By concatenating per-pair embeddings, we ensure a joint representation of all input-output pairs, allowing the decoder to access information from all grids during inference. Processing input-output pairs independently in the encoder serves as a strong prior for capturing high-level program features, reducing the risk of learning spurious correlations between pixels of different pairs. In contrast, methods that process all raw pairs jointly increase the encoder's computational demands. Additionally, feeding all raw pairs directly to the decoder would complicate positional encodings, requiring simultaneous modeling of both within-pair and across-pair positions. Our approach mitigates this by encoding positional information separately within each pair at the encoder stage. The decoder then receives each pair's embedding in a distinct position within the concatenated sequence, enabling clear differentiation between examples.

### H.2 Test-Time Fine-Tuning

We implement the following test-time parameter tuning approach where the transductive model's parameters are fine-tuned on the specification itself. Given a specification of $n$ input-output pairs, we perform gradient updates on the model parameters to better predict each output given its input and the remaining pairs.

**Update Process:** Starting from the pre-trained parameters $\theta$ and $\phi$ (decoder and encoder respectively), for each pair $(x_i, y_i)$ in the specification, we compute the loss:

$$\mathcal{L}_{\text{TTT}}^i(\phi, \theta) = -\log p_\theta(y_i|x_i, z_{\text{cat}}^{-i}) \tag{12}$$

where $z_{\text{cat}}^{-i} = [e_\phi(x_1, y_1); \ldots; e_\phi(x_{i-1}, y_{i-1}); e_\phi(x_{i+1}, y_{i+1}); \ldots; e_\phi(x_n, y_n)]$ represents the concatenated embeddings of all pairs except the $i$-th.

We then update the parameters using gradient descent:

$$\phi' = \phi - \alpha \nabla_\phi \sum_{i=1}^{n} \mathcal{L}_{\text{TTT}}^i(\phi, \theta) \tag{13}$$

$$\theta' = \theta - \alpha \nabla_\theta \sum_{i=1}^{n} \mathcal{L}_{\text{TTT}}^i(\phi, \theta) \tag{14}$$

where $\alpha$ is the learning rate for test-time adaptation.

**Inference:** After $K$ steps of parameter updates, we use the tuned parameters $\phi'$ and $\theta'$ to make predictions on new inputs. The prediction process remains the same as the transductive baseline but uses the adapted parameters:

$$\hat{y}_{n+1} \sim p_{\theta'}(y|x_{n+1}, [e_{\phi'}(x_1, y_1); \ldots; e_{\phi'}(x_n, y_n)]) \tag{15}$$

# I  ARC-AGI Approaches Summary

Comparing methodologies on the ARC-AGI benchmark requires careful consideration of the diverse variables at play. High-performing methods, such as the `o3-preview-low` model and the ARC-AGI 2024 winners, often differ substantially in their training data, test-time procedures, and computational assumptions. Our work is specifically designed to control for these variables, aiming to isolate and understand the dynamics of test-time adaptation strategies rather than to maximize performance through increased compute or specific methodological biases. To provide a comprehensive overview and situate our findings within this broader research landscape, we present a summary table outlining various prominent methodologies. The table details their model sizes, training strategies, test-time adaptation approaches, and reported performances.

Table 17: Key for Table Column Abbreviations

| Shorthand Column | Description |
|---|---|
| Lang. Pre-Train | Indicates whether the model underwent large-scale pre-training on a general language corpus. |
| ARC-AGI FT | Specifies if the model was fine-tuned on the official ARC-AGI training set. |
| Re-ARC FT | Denotes fine-tuning on the Re-ARC dataset, a procedurally generated dataset for ARC. |
| ARC Heavy FT | Refers to fine-tuning on ARC Heavy, an augmented and more difficult version of the ARC dataset. |
| CoT Infer. | Shows if the model uses Chain-of-Thought reasoning during inference. |
| Prog. Infer. | Indicates whether the model generates a program or uses a Domain-Specific Language (DSL) to solve the task at inference time. |
| Model Size | The number of parameters in the model (e.g., M for million, B for billion). |
| ARC-AGI Comps. | Notes whether the method includes components or strategies specifically designed for the ARC-AGI benchmark. |
| ARC-AGI v1 Perf. | The reported accuracy on the ARC-AGI evaluation set. |

Table 18: Comparison of Methods on ARC-AGI Benchmark (Part 1 of 2): Training and Inference Strategies

| Method | Lang. Pre-Train | ARC-AGI FT | Re-ARC FT | ARC Heavy FT | CoT Infer. | Prog. Infer. |
|---|---|---|---|---|---|---|
| `o3-preview-low` | Yes | Yes | Not public | Not public | Yes | No |
| ARChitects | Yes | Yes | Yes | Yes | No | No |
| Grok4 thinking | Yes | Not public | Not public | Not public | Yes | No |
| Grok 3 mini low | Yes | Not public | Not public | Not public | Yes | No |
| Qwen3-256b instruct | Yes | Not public | Not public | Not public | Yes | No |
| GPT4.5 | Yes | Not public | Not public | Not public | Yes | No |
| Codeit | Yes | No | Yes | No | No | Yes |
| Mirchandani | Yes | No | No | No | No | No |
| LPN | No | No | Yes | No | No | No |
| LPN + Latent Search | No | No | Yes | No | No | No |
| Inductive Li et al. | Yes | No | Yes | Yes | No | Yes |
| Transductive Li et al. | Yes | No | Yes | Yes | No | No |
| Llama 4 Maverick | Yes | Not public | Not public | Not public | Yes | No |

Table 19: Comparison of Methods on ARC-AGI Benchmark (Part 2 of 2): Model Details and Performance

| Method | Model | Model Size | ARC-AGI Comps. | ARC-AGI v1 Perf. |
|---|---|---|---|---|
| o3-preview-low | o3-preview-low | Not public | | 75.7% |
| ARChitects | Mistral-NeMo-Minitron-8B-Base | 8B | [1] | 56.0% |
| Grok4 thinking | Grok-4 | Not public | | 66.7% |
| Grok 3 mini low | Grok-3-mini | Not public | | 16.5% |
| Qwen3-256b instruct | Qwen-3-256b | 256B | | 11.0% |
| GPT4.5 | GPT4.5 | Not public | | 10.3% |
| Codeit | CodeT5 | 220M | [2] | 14.75% |
| Mirchandani | text-davinci-003 | 175B | | 6.75% |
| LPN | Vanilla Transformer | 178M | | 7.75% |
| LPN + Latent Search | Vanilla Transformer | 178M | | 15.5% |
| Inductive Li et al. | Llama3.1-8B-instruct | 8B | [2] | 38.0% |
| Transductive Li et al. | Llama3.1-8B-instruct | 8B | | 43.0% |
| Llama 4 Maverick | Llama 4 Maverick | 400B | | 4.4% |

**ARC-AGI Comps.:** Whether the methodology directly targets the ARC-AGI benchmark with specific components.

[1] ARC-specific data augmentations for training and test time, ARC-specific tokenization scheme.
[2] ARC-specific DSL.

