# OpenReview forum: "Searching Latent Program Spaces"
_NeurIPS.cc/2025/Conference — NeurIPS 2025 spotlight_

### Official Review · Reviewer_6Xrm · 2025-06-04

**Clarity:** 3
**Significance:** 2
**Originality:** 3
**Rating:** 5
**Confidence:** 4

**Summary:**

This paper introduces a new way of performing test-time adaptation for learning algorithms from few examples. The general idea is to optimize a latent "function vector" in a variational inference framework. They evaluate primarily on ARC, finding that although it is not state-of-the-art, that it is very compute efficient and do many experiments to get insight on exactly what is important for it to work.

**Questions:**

I get how your method could adapt somewhat out-of-sample because you optimize $z$ at test time, but I'm skeptical that it could compositionally generalize, because fundamentally the "program" is being "executed" by a neural network, and has to be represented by a fixed length vector. Did you ever see evidence of compositional generalization?

Did you ever try also also optimizing the parameters of the encoder/decoder during test-time optimization?

Did you ever try having the encoder take multiple examples as input?

**Ethical Concerns:**

["NO or VERY MINOR ethics concerns only"]

**Final Justification:**

The authors explained a further interesting result relating to compositionality. This makes me want to raise the score even more, so I'm thinking of this as a `5.5/6`. I'm unconcerned by our sole negative reviewer, cyKr, saying that ARC is a "narrow set of visual, grid-based reasoning tasks, which are not comprehensive enough"--ARC is a rich testbed, and essentially a composite of many diverse reasoning puzzles. (I would take the plunge and upgrade to a score of `6/6` if they showed their method working on another challenging dataset, however)

**Limitations:**

yes

**Paper Formatting Concerns:**

no concerns

**Quality:**

4

**Strengths And Weaknesses:**

Strengths:
1. Creative method that is very compute efficient relative to TTT
2. I like the connection to variational inference. Unlike a lot of work on ARC, this paper tries to connect to general ideas in machine learning, rather than engage in benchmark-hacking.
3. Rich analyses on synthetic tasks which give insight on the details of what makes the method work
4. Good description of prior work

Weaknesses:
1. There are other natural ablations and variations of your system to consider; see Questions
2. Mostly only evaluated on ARC. I strongly believe the paper should not be rejected on this basis, because ARC is rich and challenging and we still do not have open-source solutions, but it is a weakness. Note Appendix B7 has experiments on a very different synthetic dataset, which also helps address this weakness. (If accepted you should probably cover these other experiments in the main text.)
3. Nitpick: What do the performance numbers for ARC mean? Is that the number of tasks solved or the percentage of tasks solved?

---

> ### Author Rebuttal · Authors · 2025-07-30
>
> We thank the reviewer for their detailed review and appreciate their recognition of the creativity and efficiency of LPN relative to TTT.
>
> Table 3 reports ARC-AGI performance as the percentage of tasks solved. We will update the table caption to clarify this.
>
> Below, we respond to the other questions posed.
>
> **Q1. “... I'm skeptical that it could compositionally generalize, because fundamentally the "program" is being "executed" by a neural network, and has to be represented by a fixed length vector. Did you ever see evidence of compositional generalization?”**
>
> We agree that since the latent program is ultimately executed by a neural network, there are inherent limitations to the compositionality achievable in this representation. LPN’s ability to perform compositionally generalization depends on the nature of the considered generalization. We investigated a variety of types of compositional generalization and found that the search component of LPN enables generalization of type Compose-Different-Concepts and Switch-Concept-Order outlined in [1].
>
> We ran experiments combining programs from the training set in novel ways not seen during training. For example, we tested a task that required both filling black cells within objects with a specific color and cropping a grid based on a rectangle, which are two operations that were only ever seen separately during training. While the encoder alone failed to generalize (only executing the cropping task), LPN with gradient-based search was able to successfully compose the two after 500 gradient steps. This demonstrates that the smooth, learned latent space supports compositional generalization beyond what was explicitly trained.
> We agree with the reviewer that illustrating the model's generalization capabilities adds clarity, and we will include this analysis in the appendix.
>
> However, we find no evidence in our experiments to suggest that LPN is capable of generalising for length-based recurrence or applying concepts in structurally different ways (e.g., chaining programs longer than those seen during training). We will also include such negative examples in the appendix.
> As noted on line 394, future work could explore discrete latent representations, potentially enabling more robust compositionality. However, learning such representations is non-trivial and must be weighed against increased training complexity.
>
> [1] Shi, Kensen, et al. "Compositional generalization and decomposition in neural program synthesis." arXiv preprint arXiv:2204.03758 (2022).
>
> **Q2. Did you ever try also also optimizing the parameters of the encoder/decoder during test-time optimization?**
>
> We did consider the possibility of hybridizing TTT and LPN by combining LPN’s latent space search with decoder parameter updates at test time. This could help when the decoder cannot fully execute a discovered latent program.
> While this may improve performance for complex programs, it would also introduce parameter updates, diverging from LPN's core advantages of efficiency and reduced overfitting. In both the pattern and systematic sequence tasks, parameter fine-tuning degraded performance. Only on the ARC-AGI benchmark have we observed increased performance, likely due to decoder capacity limitations.
> We chose not to include this hybrid in our initial work to maintain focus on LPN's unique strength: efficiently adapting without weight updates. We hypothesize that as the diversity of programs during training increases, the need for parameter updates will diminish, reinforcing the value of search-based adaptation.
>
> **Q3. Did you ever try having the encoder take multiple examples as input?**
>
> We did explore feeding multiple input-output pairs to the encoder in early versions of LPN. However, this approach had key downsides:
> It led the encoder to overfit to the number of examples seen during training, reducing its ability to generalize across varying specification sizes (as seen in Figure 3).
> It broke the permutation invariance of the I/O pairs within the specification, which is a desirable property.
> Encoding pairs independently allows for parallel computation and faster training.
> That said, the reviewer raises a valid point: ideally, representations of individual pairs could inform one another. A promising direction for future work would be to, e.g., encode each pair independently, then apply attention-based aggregation to allow cross-pair interaction and refinement.
>
> **Additional Ablation**
>
> We also draw the reviewers’ attention to an additional ablation conducted for reviewer qA5h. This evaluates different LPN test-time inference strategies on ARC-AGI, providing further insight into the roles of the program prior and gradient-based search in driving performance.

---

> > ### Comment · Reviewer_6Xrm · 2025-08-01
> >
> > Thanks for your response! A couple things are very positive:
> >
> > 1. Evidence of compositional generalization: can train on $f$, and $g$, and then be tested on $f\circ g$
> > 2. Numbers are percentages, not absolute number of tasks solved: The system is actually doing quite well on ARC. By percentage the empirical results are good.
> >
> > I disagree strongly with our sole negative reviewer, cyKr, that ARC is a "narrow set of visual, grid-based reasoning tasks, which are not comprehensive enough".
> >
> > I advise revising to include your synthetic experiments, and make mention of these compositionality results, both within the main text.

---

> > > ### Author Response · Authors · 2025-08-05
> > >
> > > We would like to thank the reviewer once again for their detailed and thoughtful review. We fully agree that the evidence of combinatorial generalization is a particularly exciting aspect of our findings. We look forward to including both positive examples and counterexamples in the final version of the paper, and referring to them in the main text.
> > >
> > > We also appreciate the reviewer’s recognition of the strength of our ARC results. As noted, when accounting for differences in training data and compute, our outcomes reflect a high degree of generalization.
> > >
> > > In addition, we’re grateful for the reviewer’s support regarding the comprehensiveness of our experiments. We will include the synthetic non-visual results in the main section of the final version to further underscore the generality of our approach.
> > >
> > > Thank you again for your valuable feedback.

---

> > > ### Comment · Reviewer_cyKr · 2025-08-06
> > >
> > > Dear Reviewer 6Xrm,
> > >
> > > May I know the reasons why you disagree with the point that only evaluating the proposed methods on ARC-AGI is not comprehensive? In the paper, it seems that there are no specific designs for visual, grid-based reasoning tasks. Thus, the proposed method has the potential to be applied to other tasks. Yet, the authors do not evaluate it on other formats of tasks.
> > >
> > > Since you have mentioned you disagree with it, please provide justifications for your opinion. Thank you.
> > >
> > > Reviewer cyKr

---

> > > > ### Comment · Area_Chair_sCPY · 2025-08-06
> > > >
> > > > Dear Reviewer 6Xrm,
> > > >
> > > > Can you please discuss with Reviewer cyKr on the ARC-AGI evaluation?
> > > > Thanks!
> > > >
> > > > AC

---

> > > > ### Comment · Reviewer_6Xrm · 2025-08-06
> > > >
> > > > hi cyKr,
> > > >
> > > > Can you suggest another visual inductive reasoning dataset, which is not in some sense strictly dominated by ARC? I think this would be a more constructive way of helping the authors improve their paper.
> > > >
> > > > But basically I think of ARC in the year 2025 as sort of like ImageNet (in the year 2014), or the Atari suite (in the year 2016): Sure, a paper is stronger if it evaluates on another benchmark, but these are the best-in-class benchmarks of their type and era.

---

> > > > > ### Comment · Reviewer_cyKr · 2025-08-09
> > > > >
> > > > > Thank you for your response. I will take that into consideration. Indeed, for visual reasoning datasets, ARC-AGI is a good benchmark. But I am also curious about the performance of LPN on text-based tasks like Game of 24. In the original paper, the authors does not mention that their method are specifically designed for visual-relaated tasks.

---

### Official Review · Reviewer_H6z6 · 2025-06-25

**Clarity:** 2
**Significance:** 3
**Originality:** 2
**Rating:** 5
**Confidence:** 4

**Summary:**

The Latent Program Network (LPN) is neural architecture that combines symbolic program synthesis with neural networks by learning a latent space of implicit programs, which are then searched using gradient-based optimization at test time. This approach addresses the limitations of traditional program synthesis and deep learning methods (lack of consistency, adaptability, and generalization), as well as the expense and overfitting issues of Test-Time Training (TTT). LPN's key contributions include its ability to adapt to new tasks through gradient-based search in learned latent space, demonstrating improved generalization and consistency over in-context learning or parameter fine-tuning. Empirical results on the ARC-AGI benchmark and a synthetic "Pattern" task show LPN's superior performance in out-of-distribution scenarios and with varying specification sizes, outperforming both in-context learning and TTT.

**Questions:**

A few questions and improvements for the paper are:

Q: The core experiments use small models which raise question about how LPN scales to the size of state-of-the-art LLMs or more complex program spaces. Demonstrating consistent LPN benefits with larger models would validate scalability and robustness claims.

Q: How LPN methods would compare with the results presented in [1].
[1] "Combining Induction and Transduction for Abstract Reasoning" Li et al. 2024

**Ethical Concerns:**

["NO or VERY MINOR ethics concerns only"]

**Final Justification:**

The summary table and additional information adds to the paper score, since it clarifies major points in clarity and quality. The authors added clarifying table and highlighted the difference from the previous work

**Limitations:**

yes

**Quality:**

2

**Strengths And Weaknesses:**

Strengths are:
-The paper tackles program synthesis and reasoning tasks by proposing a hybrid neural-symbolic method with practical test-time adaptability
-The paper shows a combination of latent space modeling and test-time optimization for program synthesis and it cuts computation cost at test time

Weaknesses are:
-It seems the evaluations are mostly for ARC-AGI which restricts conclusions on broader generalization
-The paper doesn't compare results with winners of ARC-AGI 2024.

---

> ### Author Rebuttal · Authors · 2025-07-30
>
> We thank the reviewer for their detailed feedback and address each comment below.
>
> We acknowledge that most evaluations focus on grid-based reasoning tasks. However, we direct the reviewer’s attention to the sequence-based program tasks discussed in Section B.7, which do not rely on a 2D grid structure. These tasks further illustrate the tendency of TTT to overfit and demonstrate the consistent improvement of LPN with additional gradient steps.
>
> ## Contextualizing ARC-AGI Results
>
> Regarding comparisons with other methods on ARC-AGI, we emphasize the importance of controlling variables carefully when evaluating selected test-time adaptation methods (i.e., In-context, LPN, TTT). High-performing methods such as the o3-preview-low model or ARC-AGI 2024 winners differ significantly in training, test-time data, and computational assumptions. Our selection of the ARC-AGI benchmark for evaluation was specifically aimed at isolating and understanding the dynamics of test-time adaptation strategies, rather than maximizing performance through specific methodological biases or increased compute.
>
> However, we agree that providing context about the diverse performance levels and computational assumptions across different methods is essential. To enhance clarity, we will include a summary table outlining various methodologies, their model sizes, training strategies, test-time adaptation approaches, and respective performances.
>
> | Method                | Language Pre-Training | ARC-AGI Fine-Tune | Re-ARC Fine-Tune | ARC Heavy Fine-Tune | Chain of Thought Inference | Program-Based Inference | Model Size | ARC-AGI Components | Model                         | ARC-AGI v1 eval Performance |
> | --------------------- | --------------------- | ----------------- | ---------------- | ------------------- | -------------------------- | ----------------------- | ---------- | ------------------ | ----------------------------- | --------------------------- |
> | 03-preview-low        | Yes                   | Yes               | Not public       | Not public          | Yes                        | No                      | Not public |                    | 03-preview-low                | 75.7%                       |
> | ARChitects            | Yes                   | Yes               | Yes              | Yes                 | No                         | No                      | 8B         | [1]            | Mistral-NeMo-Minitron-8B-Base | 56.0%                       |
> | Grok4 thinking        | Yes                   | Not public        | Not public       | Not public          | Yes                        | No                      | Not public |                    | Grok-4                        | 66.7%                       |
> | Grok 3 mini low       | Yes                   | Not public        | Not public       | Not public          | Yes                        | No                      | Not public |                    | Grok-3-mini                   | 16.5%                       |
> | Qwen3-256b instruct   | Yes                   | Not public        | Not public       | Not public          | Yes                        | No                      | 256B       |                    | Qwen-3-256b                   | 11.0%                       |
> | GPT4.5                | Yes                   | Not public        | Not public       | Not public          | Yes                        | No                      | Not public |                    | GPT4.5                        | 10.3%                       |
> | Codeit                | Yes                   | No                | Yes              | No                  | No                         | Yes                     | 220M       |  [2]             | Code T5                       | 14.75%                      |
> | Mirchandani           | Yes                   | No                | No               | No                  | No                         | No                      | 175B       |                    | text-davinci-003              | 6.75%                       |
> | LPN                   | No                    | No                | Yes              | No                  | No                         | No                      | 178M       |                    | LPN Encoder-Decoder  Transformer          | 7.75%                       |
> | LPN + Latent Search   | No                    | No                | Yes              | No                  | No                         | No                      | 178M       |                    | LPN Encoder-Decoder  Transformer           | 15.5%                       |
> | Inductive Li et al    | Yes                   | No                | Yes              | Yes                 | No                         | Yes                     | 8B         | [2]             | Llama3.1-8B-instruct          | 38.0%                       |
> | Transductive Li et al | Yes                   | No                | Yes              | Yes                 | No                         | No                      | 8B         |               | Llama3.1-8B-instruct          | 43.0%                       |
> | Llama 4 Maverick      | Yes                   | Not public        | Not public       | Not public          | Yes                        | No                      | 400B       |                    | LLama 4 Maverick              | 4.4%                        |
>
> (ARC-AGI Components): whether the methodology directly targets the ARC-AGI benchmark
>
> [1] ARC-specific data augmentations for training and test time, ARC-specific tokenization scheme
>
> [2] ARC-specific DSL
>
>
>
>
> ## Answers to Questions
>
> **Q1. The core experiments use small models which raise question about how LPN scales to the size of state-of-the-art LLMs or more complex program spaces. Demonstrating consistent LPN benefits with larger models would validate scalability and robustness claims.**
>
> We first note that, although the LPN models trained for ARC-AGI might be small compared to large language models, they are still relatively large, with approximately 178M parameters. These models require pixel-perfect execution on 900-pixel grids across hundreds of programs, demanding significant capability from the neural decoder.
> Regarding scaling LPN to state-of-the-art LLMs, we agree that this is an exciting direction for future research. However, state-of-the-art LLMs typically have around 200 billion parameters, and training such models exceeds the scope of this initial study, for engineering and financial reasons. Nevertheless, our experiments demonstrate LPN's consistent advantages across different scales, from 1M parameters (pattern tasks) to 178M parameters (ARC-AGI).
>
> **Q2. How LPN methods would compare with the results presented in [1]. [1] "Combining Induction and Transduction for Abstract Reasoning" Li et al. 2024**
>
> We appreciate the reviewer’s request for comparison with Li et al. (2024), which also evaluates methods on the ARC-AGI benchmark. We have included their models in the summary table provided above, alongside different methods and assumptions. It's important to note that their work uses significantly different compute and data assumptions than ours. Specifically, their experiments rely on a pre-trained language model (LLaMA 3.1-8B-Instruct) for both induction and transduction strategies.
> Methodologically, their transduction approach aligns with our transduction baseline in that both use similar inference strategies: they do not explicitly represent programs but instead predict outputs directly from input-output examples. However, our work differs in its assumptions regarding model size and pre-training data. These distinctions are critical to contextualizing performance and understanding the broader implications of test-time adaptation methods like LPN.

---

> > ### Comment · Reviewer_H6z6 · 2025-08-01
> > **Thank for clarification**
> >
> > Thank you for the answers and clarifications. The summary table and additional information adds to the paper score, since it clarifies major points in clarity and quality.

---

> > > ### Author Response · Authors · 2025-08-05
> > >
> > > Thank you for your thoughtful feedback and for recognizing the added clarity provided by the summary table and additional information. We are confident that including the summary table in the final version will further highlight the key takeaways of our experiments and clarify how our results relate to prior work.

---

### Official Review · Reviewer_qA5h · 2025-07-01

**Clarity:** 4
**Significance:** 3
**Originality:** 4
**Rating:** 5
**Confidence:** 4

**Summary:**

This paper presents a novel approach for program search and applies it to the ARC-AGI benchmark. The approach uses a VAE to encode program synthesis tasks into a patent program-like space, then optimizes the latents using test time gradient updates. This is like transduction models with TTT but instead of optimizing the full transduction network, you just optimize the latent, which could be more efficient. Compared to the transduction baseline, the proposed model works much better. Performance on ARC is comparable to prior approaches, although it falls short of recent transductive models, possibly due to reduced training time and dataset size and quality. The authors provide thorough analysis of their method.

**Questions:**

Q1. any theories why more grad steps doesn't work?
Q2. on page 4, you say "we diverge from such transduction-based methods, as they cannot inherently adapt at test-time or ensure specification consistency" — isn't this exactly what TTT does, which transduction-based methods are compatible with?
Q3. for table 3 is "in distribution" the re-arc dataset?  is it the training set, or a validation set from RE-ARC? and then ood is ARC-AGI validation? maybe I missed this in the main text, but if it's not there it should be described there (and even better in the table too).

**Ethical Concerns:**

["NO or VERY MINOR ethics concerns only"]

**Final Justification:**

I will maintain my score. The other reviewers have largely overlapping views on the paper, except for reviewer cyKr who had low confidence in their review.

**Quality:**

4

**Strengths And Weaknesses:**

Strengths:
- The approach is really novel! It is also principled, and focused on a clear innovation. There have been many approaches tried on ARC-AGI in light of the competition last year and previous years, and this is one of the more innovative approaches.
- The paper is written well, clear, and easy to follow. The figures are beautiful (e.g. figure 1)
- The paper includes informative ablations.

Weaknesses:
- On ARC-AGI, as compute scales, the proposed approach does not work better than TTT. However, the authors do a decent job acknowledging this, and provide reasonable hypotheses for why this might be the case.
- Some of the details of how the baselines work are hard to understand in the main text. The appendix does a good job thoroughly explaining the different approaches, but I think a few more details of the baselines and how they are implemented could be useful for the main text.
- Other papers achieve much higher validation scores on ARC-AGI. Upon closer inspection, most of those higher scores come from bells and whistles and scaling, but perhaps the paper could do a stronger job explaining why its final scores aren't competitive on ARC. To this end, a table with validation scores from an increased number of approaches could be useful, just to put this paper more in context with the surrounding ARC literature (so readers don't have to do that work ourselves).
- The empirical experiments to test the benefits of the latent approach compared to TTT are not very strong. For example, in Figure 9, LPN + grad isn't that much computationally more efficient than TTT. And it's not clear to me how much the other proposed benefit, the program prior, is playing a role. Is there an experiment for this?

---

> ### Author Rebuttal · Authors · 2025-07-30
>
> We thank the reviewer for their thorough evaluation, highlighting the method's novelty, clear writing, and detailed ablations. Below, we address each question and comment provided.
>
> We note the reviewer’s suggestion to improve the explanation of baselines in the main text. We acknowledge this and will revise the main text to include sufficient details about the baselines.
>
> ## Contextualizing ARC-AGI Results
>
> The reviewer accurately notes that certain approaches achieve higher scores on the ARC-AGI benchmark but typically involve substantial, often ARC-specific additions (e.g., pre-trained and fine-tuned language models, synthetic data generation at test time). Our primary goal was to fairly compare test-time adaptation methods (In-context, TTT, and LPN) while holding other factors constant. Hence, direct comparisons with methods employing ARC-specific enhancements or additional pre-trained models would not be equitable.
>
> However, we agree that contextualizing our method's performance relative to various approaches is valuable, regardless of differing computational requirements or assumptions. We have thus compiled a comprehensive table summarizing a range of methods, pre-training and post-training strategies, and ARC-AGI scores. This provides a clearer context regarding LPN's performance relative to existing methods.
>
> [See Review H6z6 for table]
>
>
> ## Answers to questions
>
> **Q1. “any theories why more grad steps doesn't work?”**
>
> One theory is that using more gradient steps during training reduces the pressure on the encoder to produce a strong initialization for the decoder since extra gradient steps can compensate for a weaker starting point. Thus, beyond a certain threshold, the encoder may receive a less effective training signal, resulting in poorer initialization given the same computational budget. Our experiment (Figure 2) underscores the encoder's importance, indicating that weaker encoder performance might negatively impact test-time search.
> Table 1 supports this observation: training with 5 gradient steps (Grad 5) results in 0.0% encoder-only performance, whereas training with 0 steps (Grad 0) achieves 3.2%. However, with at least 5 test-time gradient steps, the Grad 5 model performs better. Grad 1 may strike the optimal balance, training a strong encoder for good initialization while still allowing effective latent space refinement through gradient steps at test time.
> To mitigate this issue, we applied progressive training, from 0 steps gradually increasing to N steps (see line 366), which ensures the encoder initially receives strong training signals. Gradient steps are only increased once the model converges, preventing encoder underfitting.
> Additionally, we hypothesize that doing more gradient steps in the latent space at test-time might eventually overfit the latent to the specification (train input-output pairs) and fail to generalize to the test input. For instance, in Table 3, there is some evidence that suggests minor overfitting occurs on the in-distribution dataset when going from 2e13 to 2e15 flops, with additional search marginally lowering performance. In the future, it would be valuable to study the extent to which specification overfitting occurs in LPN.
>
> **Q2. on page 4, you say "we diverge from such transduction-based methods, as they cannot inherently adapt at test-time or ensure specification consistency" — isn't this exactly what TTT does, which transduction-based methods are compatible with.**
>
> The reviewer makes a valid point. Our original wording intended to emphasize that transduction-based methods (e.g., TTT) typically lack an inherent mechanism for adapting to new data without modifying the model's parameters directly (unlike in-context learning and LPN). We acknowledge this was unclear and will clarify this distinction in the final paper, explicitly highlighting the benefits of methods capable of adapting to new data and maintaining consistency without relying on parameter-based fine-tuning.
>
> **Q3. for table 3 is "in distribution" the re-arc dataset? is it the training set, or a validation set from RE-ARC? and then ood is ARC-AGI validation? maybe I missed this in the main text, but if it's not there it should be described there (and even better in the table too).**
>
> The in-distribution dataset refers to the ARC-AGI training set, and the out-of-distribution (OOD) dataset is the ARC-AGI evaluation set. Neither dataset is used directly for training; instead, we use data generated by RE-ARC during training. However, RE-ARC tasks are built upon primitives from the ARC training set, making them effectively in-distribution relative to the ARC training set. While we attempt to explain this distinction in the description of Table 3, we acknowledge that it can be clearer and will provide additional clarification in the final version of the manuscript.
>
>
>
> ## Additional LPN experiments on ARC-AGI
> The reviewer notes the lack of analysis regarding the specific components contributing to LPN's performance on ARC-AGI. To address this, we conducted additional ablations exploring different test-time inference strategies to better understand the impacts of the program prior and gradient-based search. We evaluated performance on a single seed on ARC-AGI eval dataset (out of distribution relative to re-arc), keeping constant the number of sampled latents during inference, varying budgets in [10,50,100,200,400].
>
> Specifically, we compared the following inference strategies.
>
> **Encoder Sampling**:  Samples multiple latents from the encoder, evaluates the likelihood of the specification given each latent, and selects the best. This tests the value of structured, gradient-based search versus brute-force sampling near the encoder output.
>
> **Gaussian Init + Gradient Search**: Uses the same gradient-based search as core LPN but initializes from a sample drawn from the prior (Gaussian). This isolates the encoder’s role in both suggesting good initial points and improving overall performance.
>
> **Encoder Init + Local Search**: Starts from the encoder output but replaces gradient updates with a mutation operator (continuous noise perturbation). This examines the benefit of gradient-based updates versus unstructured local search, while still allowing movement beyond the encoder distribution.
>
> **Gaussian Init + Local Search**: Combines prior-based initialization with local search. This baseline helps evaluate the importance of both encoder initialization and gradient guidance.
>
> **Encoder Init + Gradient Search (LPN)**: The standard LPN inference method.
>
> Below is the table comparing these different inference strategies with varying search budgets.
>
>
> | Inference Method                   |    10 |    50 |   100 |   200 |   400 |
> |--------------------------------------|-------|-------|--------|--------|--------|
> | Encoder Sampling                     |  9.12 | 10.75 | 10.13 | 10.50 | 10.62 |
> | Gaussian Init + Gradient Search |  1.38 |  2.75 |  5.37 |  6.88 | 10.75 |
> | Encoder Init + Local Search     |  8.38 |  9.50 | 10.37 | 10.75 | 10.75 |
> | Gaussian Init + Local Search           |  0.25 |  0.25 |  0.25 |  0.75 |  0.25 |
> |  Encoder Init + Gradient Search (LPN)  |  9.10 | 13.25 | 15.00 | 15.50 | 15.50 |
>
> ### Summary of Results
>
> Repeated sampling from the encoder offers slight improvements over a single sample (7.75%), but the gains are minimal. Comparing _Gaussian Init + Gradient Search_ to _Encoder Init + Gradient Search_ highlights the encoder's importance: it not only boosts performance with few gradient steps but also significantly improves results at high budgets (e.g., 400 steps).
> _Encoder Init + Local Search_ performs similarly to gradient-based search at low budgets (10 steps), but diverges sharply as the budget increases, demonstrating the effectiveness of gradient guidance.
> _Gaussian Init + Local Search_ performs poorly, with results near 0%, showing the combined importance of good initialization (via the encoder) and gradient-based search.
> Overall, _Encoder Init + Gradient Search (LPN)_ consistently outperforms all other methods, confirming that LPN’s core components, i.e., encoder-informed initialization and gradient-based optimization, are essential for strong performance.

---

> > ### Comment · Reviewer_qA5h · 2025-08-01
> >
> > Thank you for the response. Your hypotheses and evidence explaining why more grad steps doesn't work is interesting. The ablation study is also really insightful. Great paper!

---

> > > ### Author Response · Authors · 2025-08-05
> > >
> > > We would like to sincerely thank the reviewer for their detailed and thoughtful review, as well as for the positive feedback on our work. We greatly appreciate the time and care they dedicated to the evaluation.
> > >
> > > We agree that the suggested additional ablation on ARC-AGI would further strengthen the paper’s contributions, and we look forward to incorporating it into the final version.

---

### Official Review · Reviewer_cyKr · 2025-07-02

**Clarity:** 3
**Significance:** 2
**Originality:** 2
**Rating:** 4
**Confidence:** 2

**Summary:**

This paper introduces the Latent Program Network (LPN), a neural architecture that learns a continuous space of implicit programs. Specifically, it adopts a VAE-like architecture and additionally uses latent representation optimization before decoding. LPN uses gradient-based search at test time to adapt to new tasks, a process that was shown to double its performance on out-of-distribution problems on the ARC-AGI benchmark. By building in this adaptation mechanism, LPN outperforms or matches other methods like in-context learning and test-time training in various settings.

**Questions:**

Please see my comments above.

Here I have to first apologize for not being able to provide many valuable suggestions. Because I am not familiar with the domain at all. I remember clearly that I clicked 'highly irrelevant' for this paper during bidding, but confused why I am still assigned this paper. Maybe this is because of the keyword "program synthesis"? As a result, I put my confidence score at only 2, and let's see how other reviewers review this paper.

**Ethical Concerns:**

["NO or VERY MINOR ethics concerns only"]

**Final Justification:**

After discussing with the authors and other reviewers, I find that I have some misunderstandings about the paper.

**Limitations:**

yes

**Quality:**

2

**Strengths And Weaknesses:**

Strength:
1. The code is open-sourced, which helps replication.
2. After reading the paper, I have a feeling that the LPN is a kind of VAE, yet we don't just decode the latent representations directly, but optimize the latent representation by either sampling or gradient ascent, which I think is valuable and interesting.
3. The performance on OOD setting significantly outperforms baseline methods.

Weakness:
1. All the evaluations are conducted based on a narrow set of visual, grid-based reasoning tasks, which are not comprehensive enough to demonstrate the superiority of LPN.
2. The paper's definition of a "program" is implicitly tied to transformations on 2D grids. The claim of creating a general approach to program synthesis is not fully verified. Still related to Weakness 1, since the authors only evaluate LPN on grid-based reasoning tasks, I doubt the claim of "program" in this paper.

---

> ### Author Rebuttal · Authors · 2025-07-30
>
> We thank the reviewer for their detailed review, for highlighting that our code is fully open-source for reproducibility, and for noting the strength of the out-of-distribution (OOD) performance of LPN.
>
> One highlighted strength was the recognition of LPN as a type of variational autoencoder (VAE), a point with which we fully agree. We clarify this further in Appendix D, explicitly framing LPN as a VAE and describing the latent optimization step as semi-amortised variational inference (SVI). Specifically, at test time, SVI refines the aggregate posterior for individual data points to close the amortization gap remaining after training. While we reference this discussion in line 182, we agree that explicitly stating in the main text that LPN performs variational inference would enhance clarity.
>
> To address the reviewer's concern about our experiments focusing narrowly on visual grid-like tasks, we highlight an additional experiment in Appendix B.7. Here, we explicitly compare LPN and test-time-training (TTT) on sequence-based programs without grid-based features, confirming that our method is not limited to grids. This experiment further demonstrates TTT's tendency to overfit and supports our claim that LPN effectively improves performance through latent-space gradient updates at test time.
>
> Regarding the comment that our definition of "program" implicitly focuses on transformations of 2D grids and thus may not fully support a general claim, we agree that our experiments do not show LPN learning a fully general space of all programs. Indeed, LPN intentionally leverages biases from its training distribution to efficiently represent a specific class of programs tied to a particular input-output distribution.
> We believe LPN is general in the sense that it makes minimal assumptions beyond having data in the form of N input-output pairs. Our experiments demonstrate that test-time adaptation improves performance consistently across all three tested environments and provides robust generalization to out-of-distribution tasks.
> However, we acknowledge the reviewer's concern. We will update the manuscript, in both the introduction and discussion, to clarify that our method targets program synthesis specifically within programming-by-example tasks, exploring grid-based and integer-sequence domains. Training LPN on a truly general program space would require extensive pre-training on diverse datasets and remains an exciting direction for future research, given sufficient compute resources.
>
> Regarding the reviewer’s concern about the claim that LPN represents a "program space," we agree that there are important distinctions between LPN's neural programs (latent vector plus neural decoder) and traditional code-based programs. Although both represent valid programs that map inputs to outputs, they differ notably in terms of compositionality, differentiability, and expressivity.
> Specifically, the space learned by LPN covers only a small subset of the broader space expressible through traditional code. While LPN sacrifices compositionality, it gains differentiability, enabling efficient test-time search, which is challenging for methods that generate code.
> We will explicitly clarify this trade-off in the final paper, emphasizing that LPN learns a narrow, implicit program space optimized for efficient search and strong performance within the training distribution.
>
> **Additional Ablation**
>
> We also draw the reviewers’ attention to an additional ablation conducted for reviewer qA5h. This evaluates different LPN test-time inference strategies on ARC-AGI, providing further insight into the roles of the program prior and gradient-based search in driving performance.

---

> > ### Author Response · Authors · 2025-08-05
> > **Invitation for Further Discussion**
> >
> > We would like to thank the reviewer once again for their thoughtful and constructive feedback. We sincerely appreciate the time and effort they dedicated to evaluating our work.
> >
> > We kindly invite you to review our rebuttal, where we have addressed each of your comments in detail. In particular, we would like to draw your attention to the additional ablation studies (reviewer qA5h) we included for LPN on the ARC-AGI benchmark. These further demonstrate the critical roles played by both the encoder and the gradient-based search in achieving strong performance.
> >
> > Thank you again for your valuable time and insights.

---

> > ### Comment · Reviewer_cyKr · 2025-08-09
> >
> > I thank the authors for the reply. I guess the authors have already seen my discussion with another reviewer. After discussion, I still hold the opinion that the evaluation of this paper is not comprehensive, focusing only on visual-related reasoning tasks. And the authors do not emphasize that their methods are specifically designed for visual-related tasks.
> >
> > Therefore, I suggest that the authors should either conduct experiments on text-based tasks like Game of 24, Mini crosswords (both of them from [1]), or limit the scope of the paper.
> >
> > Yet, I want to reiterate that I am not familiar with this topic. According to the reviews from other reviewers, this seems not a problem. Thus, I decided to keep my rating and low confidence. Let's see how AC will decide.
> >
> >
> > [1] Tree of Thoughts: Deliberate Problem Solving with Large Language Models

---

> ### Comment · Area_Chair_sCPY · 2025-08-06
>
> Dear reviewer,
>
> Please read the rebuttal and discuss with the authors if you have any questions. Thanks!
>
> AC

---

### Note · Authors · 2025-08-13

We thank the AC and reviewers for their thoughtful feedback and constructive discussions. We appreciate the opportunity to provide final remarks on the review process and discussions.

In response to 6Xrm, we analyzed compositionality with clear positive/negative cases, which will be added as a new section in the appendix. In addition, following feedback, we will move the non-visual, non-grid sequence experiments to the main text.

H6z6 and qA5h had no remaining concerns; qA5h highlighted ARC-AGI ablations as being very insightful for showing the value of the encoder for search initialization and gradient-based search.

For cyKr, we clarified that LPN targets PBE with N input-output pairs (not single-task benchmarks like Sudoku and Game of 24); the reviewer acknowledged this. We also clarified that LPN succeeds on non-visual sequence PBE tasks (Appendix B.7), which we will surface in the main text. We focus on ARC-AGI as the most stringent open PBE benchmark, enabling fair comparisons with in-context learning and TTT, and isolating LPN’s core novelty, i.e., gradient-based search in a learned latent program space, yielding compositional generalization, OOD gains, and search efficiency.

We appreciate the reviewers’ recognition of the method’s originality, clarity, and insightful ablations. We will incorporate all suggested clarifications into the revised version, including expanded baseline descriptions, evidence of compositionality, and broader performance context.

Best wishes,

The authors

---

### Decision · Program_Chairs · 2025-09-17

**Decision:**

Accept (spotlight)

**Comment:**

(a) Summary of Scientific Claims and Findings

The paper introduces the Latent Program Network (LPN), a novel architecture that learns a latent space of implicit programs. This allows the model to use gradient-based search at test time to adapt to new tasks, aiming to combine the adaptability of program synthesis with the scalability of deep learning. The key finding is that LPN's performance on out-of-distribution tasks doubles when this test-time search is enabled.

(b) Strengths

1. The core concept of a searchable latent program space is considered novel and interesting.

2. The paper provides strong evidence of the model's effectiveness on out-of-distribution tasks.

3. The paper is well-written and easy to follow.

(c) Weaknesses

1. The experiments are confined to a narrow set of visual, grid-based tasks, which raises questions about the method's general applicability.

2. The term "program" seems implicitly tied to the specific grid-based tasks used in the evaluation, weakening the claim of a general program synthesis approach.

(d) Most Important Reasons for Decision

The novelty of the approach and its strong out-of-distribution performance are compelling reasons to accept.

(e) Summary of Discussion and Rebuttal

The discussion centered on the paper's limited evaluation. Reviewers questioned whether the LPN could generalize to non-visual or text-based tasks. One reviewer defended the use of the ARC benchmark as "best-in-class" for visual inductive reasoning, but the core issue of the method's generalizability beyond this domain remained the primary point of contention.